# The Value of Variance: Mitigating Debate Collapse in Multi-Agent Systems via Uncertainty-Driven Policy Optimization

Luoxi Tang [1]   Yuqiao Meng [1]   Joseph Costa [1]   Yingxue Zhang [1]   Muchao Ye [2]   Zhaohan Xi [1]

## Abstract

Multi-agent debate (MAD) systems improve LLM reasoning through iterative deliberation, but remain vulnerable to *debate collapse*, a failure type where final agent decisions are compromised on erroneous reasoning. Existing methods lack principled mechanisms to detect or prevent such failures. To address this gap, we first propose a hierarchical metric that quantifies behavioral uncertainty at three levels: *intra-agent* (individual reasoning uncertainty), *inter-agent* (interactive uncertainty), and *system-level* (output uncertainty). Empirical analysis across several benchmarks reveals that our proposed uncertainty quantification reliably indicates system failures, which demonstrates the validity of using them as diagnostic metrics to indicate the system failure. Subsequently, we propose a mitigation strategy by formulating an uncertainty-driven policy optimization to penalize self-contradiction, peer conflict, and low-confidence outputs in a dynamic debating environment. Experiments demonstrate that our proposed uncertainty-driven mitigation reliably calibrates the multi-agent system by consistently improving decision accuracy while reducing system disagreement.

## 1. Introduction

Large language models (LLMs) have entered an era of collaborative intelligence, in which multi-agent debate (MAD) systems are intensively developed to iteratively refine their reasoning (Irving et al., 2018; Du et al., 2023). Unlike single-agent generation, which is highly sensitive to instance-specific errors, MAD systems establish a new paradigm leveraging diverse perspectives to cross-validate informa-

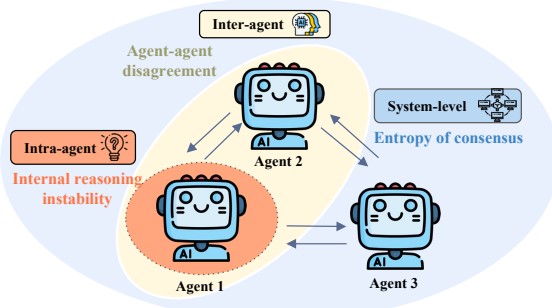

*Figure 1.* Illustration of uncertainty quantification at three levels: intra-agent (within a single LLM agent's reasoning), inter-agent (between a pair of agents during interaction), and system-level (the overall multi-agent system's decision).

tion (Perez et al., 2022), thereby demonstrating improved effectiveness and robustness across a wide range of tasks such as mathematical reasoning (Cobbe et al., 2021), factual question answering (Lin et al., 2022), and commonsense generation (Talmor et al., 2019; Zhu et al., 2024).

However, multi-agent debate is not a culminating solution. Recent studies have revealed its vulnerability to *debate collapse*. Specifically, researchers have identified that agents are influenced by a confident but compromised minority (Khan et al., 2024), or become trapped in endless loops of conflicting arguments without convergence (Cemri et al., 2026; Foerster et al., 2016). Notably, the behavioral analysis of these failures remains underexplored. Existing methods often treat debate collapse as an outcome-level error rather than a dynamic procedural failure (Du et al., 2023; Perez et al., 2022), hence lacking the granular metrics needed to identify how and when a debate begins to derail. Current mitigation strategies typically rely on external judge agents or rigid handoff interactions, which fail to address the intrinsic instability and robustness limitations of the debating agents themselves (Chan et al., 2024; Perez et al., 2022).

These limitations imply two research gaps: (1) the lack of well-established frameworks to indicate the debate collapse, and (2) the absence of mitigation that inherently improves MAD robustness, rather than temporarily patching failures with auxiliary components (e.g., introducing a judge agent).

**This work.** To address these two gaps, this paper conducts

[1]Department of Computer Science, Binghamton University, Binghamton, NY, USA [2]University of Iowa, Iowa City, IA, USA. Correspondence to: Zhaohan Xi <zxi1@binghamton.edu>.

*Proceedings of the 43rd International Conference on Machine Learning*, Seoul, South Korea. PMLR 306, 2026. Copyright 2026 by the author(s).

research at two levels:

### I. Uncertainty-driven indicators of debate collapse

Fundamentally, the failure of AI systems is frequently accompanied by observable behavioral indicators (Kapoor et al., 2024; Kuhn et al., 2023). Just as a confused human might hesitate, stutter, or contradict themselves, an AI system struggling with a task often exhibits instability in its outputs and inconsistency in its internal states, particularly when confidence is misaligned with correctness (Xiong et al., 2024; Tomani et al., 2025). In the multi-agent setting, such phenomena manifest along more complex dimensions. Prior work has shown that agent-based deliberation can suffer from coordination breakdowns, oscillatory behaviors, and failure modes induced by misaligned beliefs or unreliable reasoning processes (Du et al., 2023; Cemri et al., 2026). For example, agents that frequently revise their positions, diverge from peer models, or produce low-confidence outputs may disrupt overall system robustness. These behavioral volatilities are not merely noise; rather, they constitute intrinsic signals of the system's reliability. Intuitively, quantifying such signals enables diagnosing the robustness of a debate process before it culminates in an incorrect final decision.

Correspondingly, we propose a hierarchical framework (Figure 1) to quantify uncertainty at three behavioral levels: **(i) Intra-agent**, we measure self-consistency by tracking how often an individual agent flips its stance during the debate, motivated by evidence that frequent belief revision correlates with unreliable reasoning (Wang et al., 2023). **(ii) Inter-agent**, we quantify peer disagreement, since persistent conflict among agents often reflects ambiguity or hallucinated facts rather than productive deliberation (tse Huang et al., 2025; Cemri et al., 2026). **(iii) System-level**, we assess the entropy of the final consensus, which serves as a holistic confidence measure for the collective output and aligns with established uncertainty-based diagnostics (Kuhn et al., 2023; Tomani et al., 2025).

Using uncertainty-driven quantification, we conduct extensive empirical analyses on standard reasoning benchmarks (Cobbe et al., 2021; Lin et al., 2022; Talmor et al., 2019). Our experiments reveal that incorrect predictions consistently exhibit significantly higher intra-agent flip rates, inter-agent disagreement, and system-level output entropy than correct predictions. This demonstrates that debate collapse is not random; rather, it leaves a distinct behavioral footprint that differentiates failure cases from successful agent deliberation. These results validate our proposed indicators as reliable diagnostic tools for assessing robustness in multi-agent debate systems.

### II. Uncertainty-driven collapse mitigation

Subsequently, we use uncertainty-derived indicators to guide our mitigation design. We propose Uncertainty-Driven Policy Optimization (UDPO), which models multi-agent debate as a dynamic interaction process and introduces an uncertainty-driven reward function that penalizes self-contradiction (intra-stability), peer conflict (inter-stability), and low-confidence outputs (system-stability). To account for agents presenting different uncertainty levels, we further design an asymmetric optimization objective that assigns agent-specific penalties while incorporating a shared task reward to ensure problem-solving correctness.

Finally, we demonstrate that UDPO-based mitigation substantially enhances the resilience of multi-agent debate systems. Experimental results show that MAD fine-tuned with this objective recover from collapse scenarios with more than 25% higher accuracy than baseline multi-agent fine-tuning methods (Yu et al., 2022; He et al., 2023). Even with explicit attacks (tse Huang et al., 2025) (i.e., compromised agents) in MAD, our method effectively reduces hallucination-induced uncertainty and improves the MAD accuracy, implying a viable solution toward robust multi-agent intelligence.

In summary, this paper makes three contributions: (1) We propose hierarchical uncertainty indicators over MAD behaviors to diagnose debate collapse. (2) We empirically and statistically demonstrate that debate collapse is positively correlated with these uncertainty indicators. (3) We introduce an uncertainty-driven policy optimization (UDPO) strategy to mitigate debate collapse. Through extensive evaluations, we show that UDPO effectively reduces debate collapse and remains robust to attacks during multi-agent consensus.

## 2. Uncertainty Quantification in MAD

This section details the uncertainty-driven quantification as indicators of debating collapse.

**Setting.** We study multi-agent debate (MAD) systems in which a set of agents $\{\mathcal{A}_i\}_{i=1}^N$ engage in $T$ rounds of deliberation to answer a question $q$. We follow a representative design in Du et al. (2023); Chan et al. (2024), where agents iteratively exchange arguments, critique each other's responses, and refine their reasoning over multiple debate rounds.

We propose to quantify uncertainty at three complementary levels: (i) *intra-agent*–how much each agent changes its own mind during debate, (ii) *inter-agent*– how much agents disagree with each other, and (iii) *system-level*–how confident the final prediction is. All quantities are defined per question $q$ and later aggregated over datasets.

**Intra-agent uncertainty.** Let $a_t^{(i)} \in \mathcal{Y}$ be the answer of agent $i$ after round $t$, with $t = 0$ denoting the initial response

before any debate. We measure how often agents flip their stance during deliberation:

$$F(q) = \frac{1}{N(T-1)} \sum_{i=1}^{N} \sum_{t=1}^{T-1} \mathbf{1}\big[a_t^{(i)} \neq a_{t+1}^{(i)}\big]$$

which we call the *flip rate*. We also measure whether agents end up with a different answer than they started with:

$$M(q) = \frac{1}{N} \sum_{i=1}^{N} \mathbf{1}\big[a_0^{(i)} \neq a_T^{(i)}\big]$$

the *belief revision rate*. The intra-agent uncertainty is $U_{\text{intra}}(q) = \lambda F(q) + (1-\lambda)M(q)$ with $\lambda \in [0,1]$. Large $U_{\text{intra}}$ means agents frequently revise their answers, indicating unstable reasoning on this question.

**Inter-agent uncertainty.** In each round $t$, we measure how much agents disagree with each other via pairwise conflict:

$$C_t(q) = \frac{2}{N(N-1)} \sum_{1 \leq i < j \leq N} \mathbf{1}\big[a_t^{(i)} \neq a_t^{(j)}\big]$$

and aggregate over rounds: $U_{\text{inter}}(q) = \frac{1}{T+1} \sum_{t=0}^{T} C_t(q)$. High $U_{\text{inter}}$ means agents perform fluctuations during the debate, suggesting the question incurs large discrepancy in decision-making.

**System-level uncertainty.** Finally, we quantify system-level uncertainty by aggregating three complementary signals: (1) normalized entropy $\tilde{H}(q)$, measuring output diversity across all rounds;(2) final disagreement $D(q) \in [0,1]$, reflecting the lack of consensus at termination; and (3) leave-one-out instability $L(q)$, capturing the system's sensitivity to individual agents. The system-level uncertainty is defined as:
$$U_{\text{sys}}(q) = \frac{1}{3}\Big[\tilde{H}(q) + D(q) + L(q)\Big]$$

Formal definitions of all three components are provided in Appendix D.

**Rationale.** Our intuition is that correct predictions typically exhibit low uncertainty across these levels (agents remain stable, agree with one another, and converge confidently) whereas incorrect predictions tend to show elevated uncertainty in at least one component. Next, we empirically validate this intuition across diverse reasoning tasks.

# 3. Evaluations of Uncertainty Quantification

This section aims to empirically evaluate the usefulness of uncertainty quantification.

## 3.1. Evaluation Setting

**MAD system.** We adopt a representative MAD setup following Du et al. (2023); Chan et al. (2024), which is used in many existing MAD systems (Liang et al., 2024; Khan et al.,

2024; Wu et al., 2024). In this setup, agents operate in a flat role structure: they first answer a question independently, then iteratively observe, critique, and refine each other's responses over multiple discussion rounds, and finally reach a decision via majority vote.

**Datasets.** We evaluate on three intensively used benchmarks on different domains: (i) GSM8K (Cobbe et al., 2021): multi-step mathematical reasoning; (ii) TruthfulQA (Lin et al., 2022): factual QA on misconceptions; (iii) CommonsenseQA (Talmor et al., 2019): commonsense knowledge.

**Models.** We implement a multi-agent ($N = 5$), multi-round ($T = 5$) debate using LLMs: Llama-4-Scout, Qwen-3-8B, Gemma-3n-E4B, Mixtral-8x7B, and Phi-4. Here we select open-source LLMs to facilitate later fine-tuning. We also evaluate different number of agent groups ($N = 3$ or 10) in §5.

**Debate collapse settings.** We consider two conditions: **(1) Natural collapse (no attack):** We directly evaluate a natural MAD system. **(2) Compromised agent (with attack):** Following the AUTOTRANSFORM attack (tse Huang et al., 2025), one agent is compromised and deliberately introduces stealthy errors to subtly mislead others.

## 3.2. Evaluation Results

This section details our evaluation results. Due to space limitation,we defer complementary results to Appendix B.

**Uncertainty distributions.** Figure 2 compares uncertainty scores for failed vs. successful reasoning outcomes. Across all uncertainty types, failures consistently show higher uncertainty than successes. For example, the mean system-level uncertainty ($U_{\text{sys}}$) is 0.67 for failures, compared to 0.09 for successes. This gap is further supported by hypothesis tests yielding small $p$-values (e.g., $p < 0.001$) and large effect sizes (Cohen's $d > 0.8$) (Cohen, 2013), indicating a statistically significant and practically meaningful separation between the uncertainty distributions of failed and successful reasoning.

**Correlation analysis.** Figure 3 presents the correlation matrix. All three uncertainty measures show significant negative correlations with accuracy ($r < 0$, $p < 0.001$), indicating that higher uncertainty reliably predicts lower accuracy. Interestingly, the uncertainty measures are positively and significantly correlated with one another ($r$ approaches 1), yet their distributions differ significantly ($p < 0.001$). This suggests that they provide a consistent overall signal while capturing complementary aspects of uncertainty.

**Utility of uncertainty metrics for filtering adversarial attacks.** We also consider an attack-involved MAD setting in which some agents may be compromised. Following AU-TOTRANSFORM (tse Huang et al., 2025), we inject stealthy

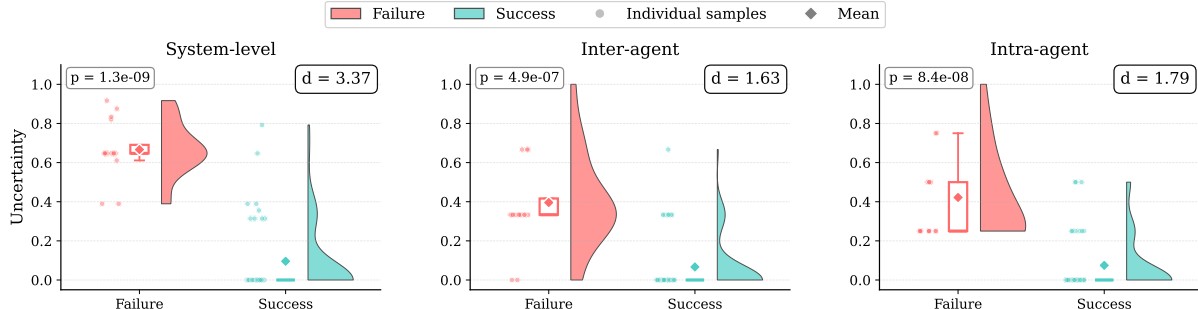

*Figure 2.* Uncertainty distributions on GSM8K for failed vs. successful reasoning. We conduct two-sample $t$-test with the null hypothesis ($H_0$) that the two distributions are identical. A smaller $p$-value (e.g., $p < 0.05$) indicates stronger evidence to reject $H_0$, suggesting that the two distributions are statistically different. We also compute Cohen's d (Cohen, 2013) to quantify the standardized difference between the means of two distributions ($d > 0.8$ indicates a large difference). Results for other datasets are shown in Appendix B.1.

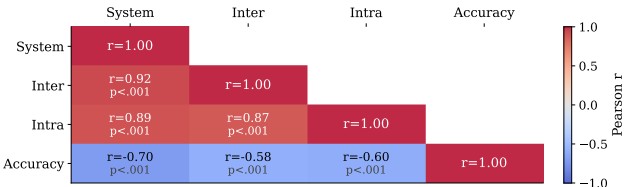

*Figure 3.* Correlation matrix (TruthfulQA) by Pearson's correlation coefficient $r$ (Wikipedia contributors, 2026) . We also conduct two-sample $t$-test same as in Figure 2. All uncertainty types negatively correlate with accuracy ($r < 0, p < 0.001$). Results for other datasets are shown in Appendix B.2.

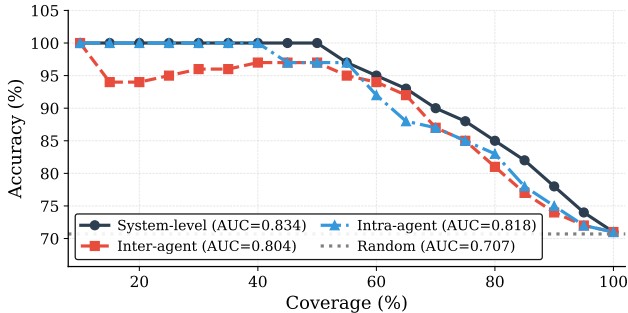

*Figure 4.* Uncertainty-driven data selection on TruthfulQA. See Appendix B.3 for additional results.

reasoning errors that remain superficially plausible but lead to incorrect conclusions. We then conduct an uncertainty-driven data selection analysis to evaluate whether uncertainty metrics can filter adversarial reasoning and mitigate attack impact. Specifically, we rank all reasoning traces by each uncertainty measure ($U_{\text{intra}}$, $U_{\text{inter}}$, and $U_{\text{sys}}$), and retain the top $k\%$ with the lowest uncertainty. Figure 4 reports accuracy as $k$ varies. We observe a clear trend: low-uncertainty subsets (e.g., $k = 50\%$) achieve substantially higher accuracy (up to 100%), demonstrating that uncertainty metrics can distinguish normal reasoning from adversarially perturbed reasoning. As $k$ increases, accuracy degrades sharply as more high-uncertainty (and often failed) instances are included. The same pattern holds across all datasets (Figure 8). These results confirm that uncertainty is an effective criterion for isolating reliable reasoning, and motivate our uncertainty-driven replay mechanism, which prioritizes high-uncertainty traces during training to reduce failures (Appendix A.3).

Due to space limitations, we defer additional analyses of uncertainty distributions and correlations to Appendix B.

**Summary.** Our experiments reveal several findings about the usefulness of uncertainty in MAD systems. **First**, all three uncertainty measures ($U_{\text{intra}}$, $U_{\text{inter}}$, and $U_{\text{sys}}$) are positively associated with system failures, confirming that un-

certainty provides meaningful signals across multiple granularities. **Second**, although these measures are strongly correlated, their distributions differ significantly, reflecting distinct failure modes in multi-agent deliberation: agents may make isolated mistakes that are later corrected (or amplified) through interaction, and a self-confident agent can still be shifted by other deliberating agents. This highlights the need for a multi-aspect view that considers all three uncertainty levels rather than collapsing them into a single score. **Third**, uncertainty metrics remain effective under adversarial perturbations, indicating that they capture robust signals of collaborative reasoning quality regardless of whether failures arise naturally or are induced by attacks.

> **Insight ♀.** Our three-level uncertainty quantification ($\overline{U_{\text{intra}}, U_{\text{inter}}}, U_{\text{sys}}$) serves as **reliable indicators of MAD system failures**, which provide actionable signals for subsequent mitigation design.

## 4. Uncertainty-Driven Mitigation

Using uncertainty metrics as indicators of debate collapse (§3), this section designs a mitigation framework to improve system robustness.

## 4.1. Overview

Given that debate collapse arises during agent interaction, we formulate our mitigation in a dynamic debating environment in which agents iteratively exchange critiques and revise their responses. In this process, we quantify uncertainty at three levels: $U_{\text{intra}}$ captures instability within an agent's reasoning trajectory, $U_{\text{inter}}$ measures disagreement between agents, and $U_{\text{sys}}$ reflects a lack of confidence in the final output. We further design an **agent-tailored, asymmetric policy optimization objective** to address agents' different uncertainty levels. Minimizing this objective will drive the debate dynamics toward robust consensus.

## 4.2. Uncertainty-Driven Reward Design

As multi-agent debate presents in a dynamic, interactive environment, we design a reinforcement-style reward driven by uncertainty metrics (§2). We treat each agent as a policy that generates responses over multiple debate rounds. The resulting multi-agent debate trajectory, $\tau = (a_0^{(1)}, \ldots, a_0^{(N)}), \ldots, (a_T^{(1)}, \ldots, a_T^{(N)})$, comprising $N$ agents across $T$ rounds, serves as the basic unit for defining reward functions.

We first define a reward that encourages intra-agent stability. Intuitively, an agent should maintain a consistent position throughout the debate unless presented with strong evidence to revise its belief:

$$r_{\text{intra}}(\tau) = 1 - \frac{1}{N(T-1)} \sum_{i=1}^{N} \sum_{t=0}^{T-1} \mathbf{1}[a_t^{(i)} \neq a_{t+1}^{(i)}] \quad (1)$$

This reward is maximized when agents exhibit stable reasoning trajectories. Stability alone is not sufficient: we incorporate task supervision in the training objective to ensure that stability is reinforced only when it aligns with correctness.

Next, we define a reward that encourages inter-agent agreement:

$$r_{\text{inter}}(\tau) = 1 - \frac{1}{T+1} \sum_{t=0}^{T} \frac{2}{N(N-1)} \sum_{i<j} \mathbf{1}[a_t^{(i)} \neq a_t^{(j)}] \quad (2)$$

This formulation captures agreement over the entire debate trajectory, rewarding early convergence over late-stage consensus.

Finally, we define a system-level reward that incorporates entropy, disagreement, and leave-one-out instability. Let $p(y \mid \tau)$ denote the empirical distribution over answers induced by the final-round responses:

$$r_{\text{sys}}(\tau) = 1 - \frac{1}{3}\left[\tilde{H}(\tau) + D(\tau) + L(\tau)\right] \quad (3)$$

Similar to Eq. 2, $\tilde{H}(\tau)$ is the normalized entropy, $D(\tau) \in \{0, 1\}$ indicates residual disagreement, and $L(\tau)$ measures leave-one-out instability, i.e., the fraction of agents whose absence will change the final decision. This reward favors confident predictions that are robust to individual agent perturbations. Note that formal definitions of each component are shown in Appendix D.

## 4.3. Asymmetric Objective for Uncertainty-Driven Policy Optimization (UDPO)

A critical insight of our approach is that **not all agents contribute equally to debate collapse**. Some agents exhibit higher uncertainty or are more susceptible to adversarial influence, while others serve as stabilizing participants. We therefore adopt an **asymmetric optimization** strategy that applies different training intensities to different agents based on their individual uncertainty profiles.

Given a debate trajectory $\tau$, the total reward for agent $i$ is:

$$r_{\text{UDPO}}^{(i)}(\tau) = \underbrace{\alpha^{(i)} r_{\text{intra}} + \beta^{(i)} r_{\text{inter}} + \gamma^{(i)} r_{\text{sys}}}_{\text{uncertainty-driven reward}} + \underbrace{\lambda^{(i)} r_{\text{task}}}_{\text{task reward}} \quad (4)$$

where $r_{\text{task}}(\tau) = \mathbf{1}[\hat{y}(\tau) = y^*]$ indicates whether the final majority-vote prediction matches the ground truth. Crucially, the coefficients $\alpha^{(i)}$, $\beta^{(i)}$, $\gamma^{(i)}$, and $\lambda^{(i)}$ are **agent-tailored**: agents with higher historical uncertainty receive stronger uncertainty-driven rewards to encourage stabilization, while already-stable agents receive relatively higher task rewards. This asymmetric weighting prevents over-regularization of well-behaved agents while focusing optimization effort where it is most needed.

**Clipped relative-update objective.** To optimize agent policies under this reward, we adopt a clipped relative-update objective that stabilizes learning by constraining the magnitude of policy changes while anchoring updates to a reference policy. Let $\pi_\theta^{(i)}$ denote the current policy for agent $i$ and $\pi_{\theta_{\text{ref}}}^{(i)}$ a fixed reference policy (e.g., a lagged snapshot or the base model). We define the trajectory-level likelihood ratio:

$$\rho_\theta^{(i)}(\tau) = \exp\left(\log \pi_\theta^{(i)}(\tau) - \log \pi_{\theta_{\text{ref}}}^{(i)}(\tau)\right) \quad (5)$$

The optimization objective for agent $i$ is:

$$\mathcal{L}_{\text{UDPO}}^{(i)}(\theta) = \mathbb{E}_\tau\left[\ell_{\text{clip}}^{(i)}\right] - \eta^{(i)} \mathbb{E}_\tau\left[\ell_{\text{anchor}}^{(i)}\right] \quad (6)$$

where expectations are over $\tau \sim \pi_{\theta_{\text{ref}}}$, $\epsilon$ controls the maximum relative update size, and $\eta^{(i)}$ is an agent-tailored anchoring weight. Agents prone to instability receive stronger anchoring ($\eta^{(i)}$ larger) to prevent erratic updates.

The clipped surrogate and anchoring terms are:

$$\ell_{\text{clip}}^{(i)} = \min\left(\rho^{(i)} \hat{A}^{(i)}, \text{clip}(\rho^{(i)}, 1-\epsilon, 1+\epsilon)\hat{A}^{(i)}\right) \quad (7)$$

$$\ell_{\text{anchor}}^{(i)} = \text{KL}\left(\pi_\theta^{(i)}(\cdot|\tau) \,\|\, \pi_{\theta_{\text{ref}}}^{(i)}(\cdot|\tau)\right) \quad (8)$$

*Table 1.* **Main results across different datasets, MAD systems, and baseline comparisons** on natural collapse (no-attack) settings. We report accuracy (%) and uncertainty metrics. Best results are highlighted by **bold** texts. $U_{\text{in}}$, $U_{\text{ir}}$, $U_{\text{s}}$ denote $U_{\text{intra}}$, $U_{\text{inter}}$, $U_{\text{sys}}$ respectively. More results under different agent selections or different numbers of debate rounds ($T$) are reported in C.1 and C.2.

| Dataset | $N$ Agents in MAD | Standard MAD | | | | MAPPO | | | | RMAAC | | | | UDPO (Ours) | | | |
|---|---|---|---|---|---|---|---|---|---|---|---|---|---|---|---|---|---|
| | | Acc | $U_{\text{in}}$ | $U_{\text{ir}}$ | $U_{\text{s}}$ | Acc | $U_{\text{in}}$ | $U_{\text{ir}}$ | $U_{\text{s}}$ | Acc | $U_{\text{in}}$ | $U_{\text{ir}}$ | $U_{\text{s}}$ | Acc | $U_{\text{in}}$ | $U_{\text{ir}}$ | $U_{\text{s}}$ |
| GSM8K | $N$=3 | 51.2 | .231 | .268 | .372 | 64.8 | .218 | .185 | .278 | 66.3 | .205 | .172 | .258 | **84.6** | **.068** | **.052** | **.078** |
| | $N$=5 | 68.4 | .228 | .172 | .285 | 73.6 | .185 | .145 | .232 | 75.8 | .168 | .132 | .215 | **92.3** | **.065** | **.038** | **.058** |
| | $N$=10 | 65.7 | .235 | .188 | .298 | 70.2 | .192 | .158 | .248 | 72.1 | .175 | .145 | .228 | **89.8** | **.071** | **.055** | **.072** |
| TruthfulQA | $N$=3 | 62.4 | .225 | .178 | .275 | 68.5 | .192 | .152 | .238 | 70.2 | .185 | .148 | .225 | **85.2** | **.072** | **.058** | **.082** |
| | $N$=5 | 71.8 | .218 | .142 | .235 | 74.2 | .158 | .122 | .198 | 76.5 | .152 | .115 | .188 | **88.7** | **.068** | **.048** | **.068** |
| | $N$=10 | 73.5 | .222 | .128 | .218 | 76.8 | .165 | .108 | .182 | 78.2 | .158 | .102 | .172 | **91.4** | **.075** | **.035** | **.055** |
| CSQA | $N$=3 | 68.2 | .205 | .162 | .258 | 72.5 | .172 | .138 | .222 | 74.1 | .168 | .132 | .212 | **86.8** | **.062** | **.048** | **.072** |
| | $N$=5 | 75.4 | .198 | .135 | .225 | 78.2 | .148 | .112 | .188 | 79.8 | .142 | .108 | .178 | **91.5** | **.058** | **.032** | **.052** |
| | $N$=10 | 73.8 | .202 | .148 | .238 | 76.5 | .155 | .125 | .202 | 77.2 | .148 | .118 | .192 | **88.2** | **.064** | **.045** | **.065** |

where we abbreviate $\rho^{(i)} = \rho_\theta^{(i)}(\tau)$ and $\hat{A}^{(i)} = \hat{A}^{(i)}(\tau)$.

The advantage estimate is computed from the agent-specific reward:

$$\hat{A}^{(i)}(\tau) = r^{(i)}(\tau) - b \qquad (9)$$

where $b$ is a batch-wise baseline used to reduce variance.

**Agent-tailored hyperparameter selection.** The coefficients $\alpha^{(i)}$, $\beta^{(i)}$, $\gamma^{(i)}$, $\lambda^{(i)}$, and $\eta^{(i)}$ are selected based on each agent's empirical uncertainty profile measured during a warm-up phase. Specifically, we compute the average $U_{\text{intra}}$, $U_{\text{inter}}$, and contribution to $U_{\text{sys}}$ for each agent over its reasoning trajectories: agents exhibiting higher uncertainty receive proportionally larger coefficients. This agent-tailored calibration ensures that the asymmetric optimization adapts to the heterogeneous behaviors of different LLMs in the debate ensemble. Full details of the parameter selection procedure are provided in Appendix A.2.

## 5. Evaluations of Mitigation

This section evaluates the UDPO-based mitigation through four research questions: (RQ1) Does UDPO improve overall performance? (RQ2) How does each loss component contribute? (RQ3) Can our method maintain effectiveness under attacks? (RQ4) When does our method help most?

### 5.1. Setup

**Datasets and models.** We use the same datasets and LLMs as Section 3.

**Baselines.** We compare against three methods. Standard MAD is vanilla multi-agent debate without training (Du et al., 2023; Liang et al., 2024). MAPPO (Yu et al., 2022) extends PPO to cooperative multi-agent settings with a centralized value function; we adapt it to MAD by treating each agent's answer selection as a policy and optimizing for consensus. RMAAC (He et al., 2023) is a robust multi-agent

actor-critic method that models state uncertainty through adversarial perturbations; we adapt it by treating answer distributions as states and training agents against perturbations.

### 5.2. Main Results (RQ1)

**Observations.** Table 1 compares all methods across dataset–model combinations. UDPO consistently achieves the best overall performance. For instance, relative to Standard MAD, UDPO yields gains of up to 25 percentage points in the $N$=5 setting (68.4% → 92.3% on GSM8K). Compared to a representative multi-agent RL baseline such as MAPPO, UDPO improves accuracy by up to 20 points (73.6% → 92.3% on GSM8K). UDPO also substantially lowers uncertainty: on GSM8K, it reduces $U_{\text{sys}}$ by roughly 80% compared to Standard MAD. This aligns with our empirical finding that uncertainty is a reliable signal of debate collapse, and that explicitly controlling it improves MAD performance.

**Explanations.** The gains largely come from *what* each training objective explicitly optimizes. **MAPPO** is essentially PPO with a *centralized value function* (critic) that can take global information during training, wherein each agent conditions only on its local observation at execution time (i.e., CTDE) (Rashid et al., 2020; Wen et al., 2021). This centralized critic helps agents coordinate under a shared reward, but it does not directly encode behavioral stability (e.g., discouraging flip-flopping across rounds or resolving persistent disagreement). As a result, agents can still reach a brittle agreement, including consensus on an incorrect rationale. On the other hand, **RMAAC** targets robustness from a different angle: it formulates MARL as a Markov game and learns policies that are robust to worst-case (potentially adversarial) state perturbations via an equilibrium learning. Even though RMAAC emphasizes robustness to corrupted observations, it is not designed to diagnose *which* debate behaviors are failing (self-contradiction vs. peer conflict vs. low-confidence aggregation) on granular agent behaviors.

*Table 2.* **Ablation study** by removing each reward component. Red highlights the most affected metric for each ablation.

| Method | GSM8K | | | | TruthfulQA | | | | CSQA | | | |
|---|---|---|---|---|---|---|---|---|---|---|---|---|
| | Acc | $U_{intra}$ | $U_{inter}$ | $U_{sys}$ | Acc | $U_{intra}$ | $U_{inter}$ | $U_{sys}$ | Acc | $U_{intra}$ | $U_{inter}$ | $U_{sys}$ |
| Full UDPO | **92.3** | **.065** | **.038** | **.058** | **88.7** | **.068** | **.048** | **.068** | **91.5** | **.058** | **.032** | **.052** |
| w/o $\mathcal{L}_{intra}$ | 86.5 | .142 | .052 | .085 | 83.2 | .138 | .062 | .092 | 85.8 | .145 | .048 | .082 |
| w/o $\mathcal{L}_{inter}$ | 88.2 | .078 | .098 | .095 | 85.4 | .082 | .105 | .098 | 87.2 | .072 | .095 | .088 |
| w/o $\mathcal{L}_{sys}$ | 84.8 | .082 | .058 | .128 | 81.5 | .085 | .068 | .135 | 83.6 | .078 | .055 | .125 |
| Standard MAD | 68.4 | .228 | .172 | .285 | 71.8 | .218 | .142 | .235 | 75.4 | .198 | .135 | .225 |

*Table 3.* **Results under attacks with multiple compromised agents.** We inject stealthy semantic errors using AUTOTRANSFORM (tse Huang et al., 2025) into $m \in \{1, 2, 3\}$ agents in an $N{=}5$ debate. We report accuracy (%) and uncertainty metrics. Best results in each row (fixed dataset and $m$) are highlighted in **bold**. $U_{in}$, $U_{ir}$, $U_s$ denote $U_{intra}$, $U_{inter}$, and $U_{sys}$, respectively.

| Dataset | # Comp. | Standard MAD | | | | MAPPO | | | | RMAAC | | | | UDPO (Ours) | | | |
|---|---|---|---|---|---|---|---|---|---|---|---|---|---|---|---|---|---|
| | | Acc | $U_{in}$ | $U_{ir}$ | $U_s$ | Acc | $U_{in}$ | $U_{ir}$ | $U_s$ | Acc | $U_{in}$ | $U_{ir}$ | $U_s$ | Acc | $U_{in}$ | $U_{ir}$ | $U_s$ |
| GSM8K | 1 | 54.9 | .286 | .255 | .522 | 62.7 | .303 | .202 | .338 | 65.4 | .232 | .188 | .315 | **79.6** | **.112** | **.084** | **.125** |
| | 2 | 41.8 | .412 | .408 | .548 | 50.6 | .318 | .275 | .468 | 54.2 | .295 | .255 | .432 | **68.9** | **.172** | **.138** | **.198** |
| | 3 | 9.4 | .662 | .525 | .614 | 36.7 | .402 | .355 | .592 | 40.5 | .372 | .332 | .548 | **56.2** | **.258** | **.212** | **.288** |
| TruthfulQA | 1 | 58.7 | .247 | .212 | .362 | 63.5 | .225 | .178 | .305 | 66.1 | .212 | .165 | .285 | **82.4** | **.108** | **.076** | **.118** |
| | 2 | 45.9 | .385 | .298 | .434 | 52.4 | .298 | .245 | .418 | 56.3 | .275 | .228 | .451 | **71.8** | **.165** | **.118** | **.182** |
| | 3 | 23.8 | .593 | .582 | .576 | 51.2 | .372 | .315 | .528 | 53.1 | .342 | .295 | .470 | **70.7** | **.242** | **.185** | **.255** |
| CSQA | 1 | 63.2 | .310 | .205 | .338 | 58.0 | .215 | .172 | .288 | 69.5 | .205 | .162 | .275 | **85.1** | **.098** | **.072** | **.108** |
| | 2 | 51.6 | .335 | .275 | .442 | 56.9 | .275 | .228 | .378 | 59.8 | .258 | .215 | .355 | **76.4** | **.148** | **.112** | **.162** |
| | 3 | 29.7 | .512 | .552 | .573 | 44.8 | .342 | .295 | .472 | 48.6 | .318 | .278 | .438 | **73.9** | **.215** | **.168** | **.235** |

In contrast, **UDPO** directly decomposes debate collapse into three interpretable failure modes and penalizes each one: intra-agent flip-flopping ($\mathcal{L}_{intra}$), inter-agent unresolved disagreement ($\mathcal{L}_{inter}$), and low-confidence final outcomes ($\mathcal{L}_{sys}$). UDPO more reliably drives debates toward stable, verifiable agreement (even in the presence of attacks) by directly optimizing against uncertainty indicators, instead of relying on centralized credit assignment (MAPPO) or worst-case perturbation robustness (RMAAC).

### 5.3. Ablation Study (RQ2)

**Observations.** Table 2 shows ablation results. Removing any component hurts both accuracy and uncertainty. The effect is targeted: removing $\mathcal{L}_{intra}$ causes $U_{intra}$ to roughly double while affecting other metrics less; similarly for $\mathcal{L}_{inter}$ and $\mathcal{L}_{sys}$. Among the three, removing $\mathcal{L}_{sys}$ causes the largest accuracy drop, followed by $\mathcal{L}_{intra}$ and $\mathcal{L}_{inter}$. This ranking is consistent across all three models.

**Insights.** Each loss component targets its intended failure mode, validating our framework design. The ranking ($\mathcal{L}_{sys} > \mathcal{L}_{intra} > \mathcal{L}_{inter}$) suggests system-level stability matters most for final accuracy (Guo et al., 2017) as it aggregates signals across all agents and rounds, capturing global patterns that local metrics miss. However, all three are necessary as we demonstrated in §3 since they represent diverse MAD aspects. The full model outperforms any ablation by a significant margin, indicating the failure modes are complementary rather than redundant.

### 5.4. Robustness to Attack (RQ3)

**Observations.** In line with stress test, Table 3 presents results when applying attacks with the varying number of compromised agents. When only one agent is attacked, standard MAD already degrades noticeably, while UDPO retains strong accuracy with only modest uncertainty growth, suggesting it can absorb localized corruption without destabilizing the debate. As we move to two and three compromised agents, all methods deteriorate, but the trend differs. Specifically, baselines exhibit a sharp escalation in $U_s$ and $U_{ir}$, indicating that attacks increasingly propagate from local inconsistencies into group-level disagreement and ultimately system-level indecision. In contrast, UDPO degrades more gracefully and maintains a larger performance margin even at higher attack rates, implying that its stability control delays the tipping point where compromised content dominates the collective outcome.

**Explanation.** The critical mechanism is not simply that UDPO improves accuracy, but that it changes how debates fail under adversarial pressure. In standard MAD (and coordination-centric baselines), an attacked agent can inject plausible but subtly flawed rationales that other agents may overfit to create a brittle consensus or a longitudinal conflict. The system then collapses between incorrect agreement and unresolved disagreement, which presents as rising $U_{in}$ and $U_{ir}$ and culminates in high $U_s$. UDPO explicitly regularizes these trends by penalizing self-contradiction, unresolved peer conflict, and low-confidence aggregation that impede

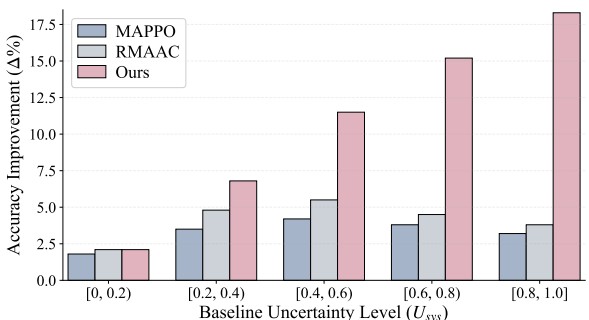

*Figure 5.* **Accuracy improvement ($\Delta\%$) over Standard MAD by uncertainty level.** Results averaged across datasets and models. Our method shows increasing gains with higher uncertainty, while MAPPO and RMAAC plateau or degrade. See Appendix C.4 for $U_{\text{intra}}$- and $U_{\text{inter}}$-based difficulty stratification.

consensus. To converge, the group must reduce instability signals rather than merely align on a superficial answer. Even with attacked trajectories, malicious content must now both persuade and remain stable under cross-examination, which is substantially harder. Hence, UDPO can achieve better robustness with delayed collapse even with a dominant fraction (more than half) of compromised agents.

**Appendix C.3** reports statistical analyses (similar to §3) of UDPO's mitigation effects under attack settings.

### 5.5. Analysis: When does UDPO help most? (RQ4)

To assess the sensitivity of our mitigation, we stratify questions into five difficulty levels based on the baseline system-level uncertainty, $U_{\text{sys}}$, computed under Standard MAD. This allows us to identify which uncertainty (difficulty) levels yield the largest gains from our method.

**Observations.** Figure 5 shows an accuracy improvement over Standard MAD across five uncertainty levels, wherein our method exhibits a clear monotonic trend: gains increase from +2.1% at the lowest uncertainty level ($U_{\text{sys}} < 0.2$) to +18.3% at the highest ($U_{\text{sys}} \geq 0.8$). In contrast, MAPPO shows modest gains that peak at +4.2% for medium uncertainty and drop to +3.2% for hard questions. RMAAC follows a similar pattern, peaking at +5.5% and declining to +3.8%. Neither baseline maintains effectiveness as difficulty increases.

**Insights.** The divergent patterns reveal fundamental differences in how methods handle uncertainty. MAPPO's centralized critic helps coordination when agents have moderate disagreement but cannot prevent the cascading instability that occurs in hard questions (Wang et al., 2024; Estornell & Liu, 2024). RMAAC's adversarial robustness helps when perturbations are bounded but struggles when debate dynamics become highly chaotic. Our method explicitly penalizes flip-flopping, disagreement, and low confidence,

which are exact behaviors that dominate hard questions. The increasing gains confirm that decomposing uncertainty into interpretable components is more effective than treating it as coordination failure (MAPPO) or adversarial noise (RMAAC). This aligns with findings that reasoning consistency is crucial for complex problems (Wang et al., 2023).

## 6. Related Work

**Multi-agent debate.** Multi-agent debate (MAD) has emerged as an effective approach to improve LLM reasoning by leveraging diverse perspectives and iterative refinement (Du et al., 2023; Liang et al., 2024; Chan et al., 2024). Recent work has explored various debate architectures, including structured argumentation (Khan et al., 2024), role-based discussions (Wu et al., 2024), and judge-mediated deliberation (Koutcheme et al., 2024). While these methods demonstrate improved accuracy over single-agent baselines, they primarily focus on architectural innovations and lack principled mechanisms to detect or prevent debate collapse. Our work complements this line of research by providing uncertainty metrics that diagnose system reliability and a training objective that explicitly mitigates failure modes.

**Uncertainty quantification in LLMs.** Uncertainty estimation for LLMs has been studied through various lenses, including token-level entropy (Kapoor et al., 2024), semantic equivalence clustering (Kuhn et al., 2023), and self-consistency measures (Wang et al., 2023). Recent work has also explored confidence calibration (Tian et al., 2023) and verbalized uncertainty (Xiong et al., 2024). However, these methods focus on single-agent settings and do not capture the rich behavioral dynamics present in multi-agent systems. Our hierarchical framework extends uncertainty quantification to MAD by decomposing system uncertainty into intra-agent, inter-agent, and system-level components, each capturing distinct failure signatures.

## 7. Conclusion

This paper introduces a hierarchical uncertainty quantification framework for multi-agent debate systems, capturing behavioral instability at the intra-agent, inter-agent, and system levels. We showed that these uncertainty signals consistently distinguish successful debates from failed ones across multiple reasoning benchmarks. Building on this analysis, we design UDPO, an uncertainty-driven policy optimization framework with an asymmetric training objective that encourages stable reasoning, agent agreement, and confident outputs. Empirical results demonstrate that this approach substantially improves accuracy while significantly reducing behavioral volatility, leading to more robust multi-agent systems that are both more reliable and better calibrated.

## Impact Statement

This paper presents methodological advances in multi-agent learning by introducing uncertainty-driven diagnostics and optimization objectives for improving the robustness of multi-agent debate systems. The proposed framework is intended to support research on reliable and interpretable collaborative reasoning, rather than to enable direct deployment in real-world decision-making settings. As with many general-purpose machine learning methods, the techniques studied here may be adapted to a wide range of applications, and their societal impact will depend on how they are ultimately used. We encourage future work to further investigate responsible deployment practices and human oversight when applying multi-agent learning systems in practical scenarios.

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

# A. Training Details

This appendix provides comprehensive details on training procedures, hyperparameter selection, and the uncertainty-driven replay mechanism.

## A.1. Experimental Settings

We fine-tune all models for 10 epochs using AdamW optimizer with learning rate $1 \times 10^{-5}$. Training is conducted on 8 NVIDIA A100 GPUs with a batch size of 32 trajectories per update. We use a linear warmup schedule for the first 500 steps followed by cosine decay. Gradient clipping is applied with a maximum norm of 1.0.

## A.2. Asymmetric Training and Agent-Tailored Hyper-parameter Selection

As described in Section 4.3, the reward coefficients in Eq. (4) are **agent-tailored**: different agents receive different coefficient values based on their individual uncertainty profiles. This asymmetric design recognizes that agents in a heterogeneous LLM ensemble exhibit different stability characteristics. We calibrate training intensity based on each agent's uncertainty level.

**Two-Phase Calibration Procedure.**

*Phase 1: Uncertainty Profiling.* Before training, we run each agent through a warm-up phase on a held-out subset of the training data (10% of trajectories). For each agent $i$, we compute:

- Average intra-agent uncertainty: $\bar{U}_{\text{intra}}^{(i)}$

- Average contribution to inter-agent uncertainty: $\bar{U}_{\text{inter}}^{(i)}$

- Average contribution to leave-one-out instability: $\bar{L}^{(i)} = \frac{1}{|\mathcal{D}_{\text{warmup}}|} \sum_\tau \mathbf{1}[\hat{y}(\tau) \neq \hat{y}(\tau^{-i})]$

*Phase 2: Coefficient Calibration.* We set agent-tailored coefficients proportional to the uncertainty profile:

$$\alpha^{(i)} = \alpha_{\text{base}} \cdot \left(1 + \kappa \cdot \bar{U}_{\text{intra}}^{(i)}\right) \tag{10}$$

$$\beta^{(i)} = \beta_{\text{base}} \cdot \left(1 + \kappa \cdot \bar{U}_{\text{inter}}^{(i)}\right) \tag{11}$$

$$\gamma^{(i)} = \gamma_{\text{base}} \cdot \left(1 + \kappa \cdot \bar{L}^{(i)}\right) \tag{12}$$

$$\lambda^{(i)} = \lambda_{\text{base}} / \left(1 + \kappa \cdot \bar{U}_{\text{sys}}^{(i)}\right) \tag{13}$$

$$\eta^{(i)} = \eta_{\text{base}} \cdot \left(1 + \kappa \cdot \bar{U}_{\text{sys}}^{(i)}\right) \tag{14}$$

where $\kappa > 0$ is a scaling factor controlling the degree of asymmetry. Agents with higher uncertainty receive stronger equilibrium rewards (larger $\alpha^{(i)}$, $\beta^{(i)}$, $\gamma^{(i)}$), weaker task rewards (smaller $\lambda^{(i)}$), and stronger anchoring (larger $\eta^{(i)}$).

**Base Hyper-parameters.** We use $\kappa = 1.5$ by default. The base coefficients $\alpha_{\text{base}}$, $\beta_{\text{base}}$, $\gamma_{\text{base}}$ are set according to the Cohen's $d$ normalization described above, with $\lambda_{\text{base}} = 1$ and $\eta_{\text{base}} = 0.01$.

**Rationale.** This asymmetric design serves two purposes: (1) it prevents over-regularization of already-stable agents, which could harm their task performance; and (2) it focuses training effort on agents that contribute most to debate collapse as well as improving sample efficiency.

## A.3. Uncertainty-Driven Replay

The selective prediction analysis in Section 3 demonstrates that high-uncertainty trajectories correspond to intrinsically difficult questions. Rather than discarding such samples at inference time, we leverage this signal during training by replaying high-uncertainty trajectories more frequently.

We maintain a replay buffer $\mathcal{B}$ that stores recent debate trajectories. After each training epoch, we compute a composite uncertainty score:

$$U(\tau) = \alpha \cdot (1 - r_{\text{intra}}(\tau)) + \beta \cdot (1 - r_{\text{inter}}(\tau)) + \gamma \cdot (1 - r_{\text{sys}}(\tau)), \tag{15}$$

and sample trajectories with probability $p(\tau) \propto U(\tau)^\eta$, where $\eta \geq 0$ controls prioritization strength ($\eta = 1$ by default).

We periodically refresh the replay buffer by re-evaluating stored questions with updated policies, and apply importance sampling corrections to ensure unbiased optimization.

### A.4. UDPO Algorithm

Below we detail the UDPO algorithm:

---

**Algorithm 1** Uncertainty-Driven Policy Optimization (UDPO)

---

**Require:** Number of agents $N$; debate rounds $T$; reward weights $(\alpha, \beta, \gamma, \lambda)$; clipping parameter $\epsilon$; reference regularization coefficient $\eta$; number of training iterations $K$.

1: Initialize agent policy parameters $\theta$
2: Initialize reference policy parameters: $\theta_{\text{ref}} \leftarrow \theta$
3: **for** $k = 1$ **to** $K$ **do**
4:     Sample a minibatch of tasks $\{x_m\}_{m=1}^{M}$
5:     **for all** $x_m$ **do**
6:         Run multi-agent debate with policy $\pi_\theta$ for $T$ rounds
7:         Collect trajectory:
8:             $\tau_m = \{(a_0^{(1)}, \ldots, a_0^{(N)}), \ldots, (a_T^{(1)}, \ldots, a_T^{(N)})\}$
9:         Compute uncertainty-based rewards:
10:          $r_{\text{intra}}(\tau_m), \; r_{\text{inter}}(\tau_m), \; r_{\text{sys}}(\tau_m)$
11:         Compute task reward $r_{\text{task}}(\tau_m)$
12:         Compute total reward $r(\tau_m)$ via Eq. (4)
13:     **end for**
14:     Compute baseline $b \leftarrow \frac{1}{M} \sum_{m=1}^{M} r(\tau_m)$
15:     Compute advantages $\hat{A}(\tau_m) \leftarrow r(\tau_m) - b$
16:     Compute likelihood ratios:
17:         $\rho_\theta(\tau_m) \leftarrow \exp\Big( \log \pi_\theta(\tau_m) - \log \pi_{\theta_{\text{ref}}}(\tau_m) \Big)$
18:     Update policy parameters $\theta$ by maximizing $\mathcal{L}_{\text{UDPO}}(\theta)$ (Eq. 6)
19:     Periodically update reference policy: $\theta_{\text{ref}} \leftarrow \theta$
20: **end for**
21: **return** optimized policy parameters $\theta$

---

# B. Additional Results on Uncertainty Quantification

This appendix presents supplementary experimental results on uncertainty quantification that complement the main findings in Section 3. These results focus on validating our uncertainty metrics as diagnostic tools for debate collapse, evaluated on the **natural MAD system without mitigation**.

### B.1. Uncertainty Distributions

Figure 6 extends the uncertainty distribution analysis (Figure 2 in the main text) to TruthfulQA and CommonsenseQA. Consistent with the GSM8K results, incorrect predictions exhibit significantly higher uncertainty across all three metrics. The effect sizes (Cohen's $d$) remain substantial, confirming that our uncertainty quantification reliably distinguishes failed from successful reasoning across diverse task domains.

### B.2. Correlation Analysis

Figure 7 presents correlation matrices for GSM8K and CommonsenseQA, supplementing the TruthfulQA results in the main text. All uncertainty types maintain significant negative correlations with accuracy across datasets, while being positively correlated with each other. This pattern validates that our three-level uncertainty framework captures complementary yet coherent signals of system reliability.

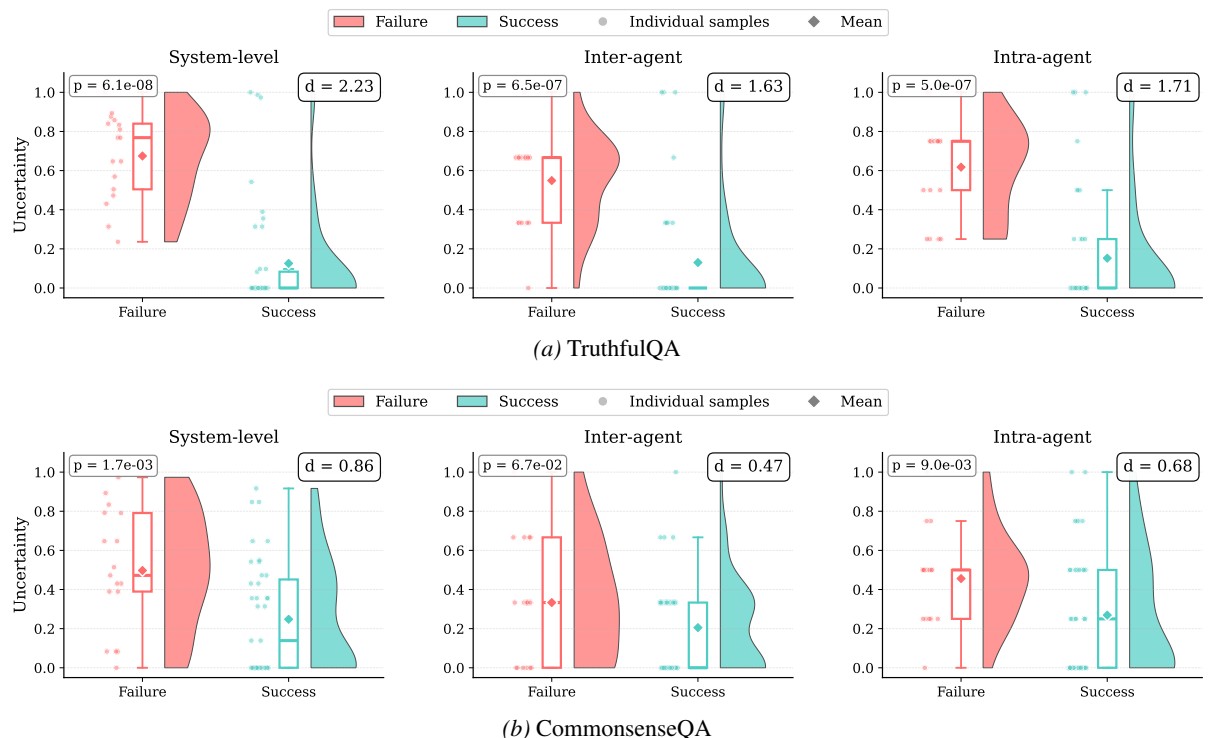

*(a)* TruthfulQA

*(b)* CommonsenseQA

*Figure 6.* **Uncertainty distributions (additional datasets).** Supplements Figure 2 in the main text. Consistent with GSM8K, incorrect predictions exhibit higher uncertainty across all metrics.

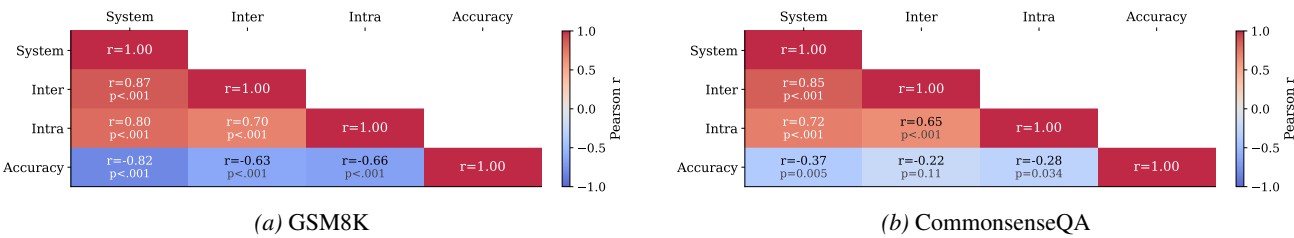

*(a)* GSM8K

*(b)* CommonsenseQA

*Figure 7.* **Correlation matrices (additional datasets).** Supplements Figure 3 in the main text. All uncertainty types negatively correlate with accuracy.

### B.3. Selective Prediction

Figure 8 demonstrates the utility of uncertainty-driven data selection on GSM8K and CommonsenseQA. The results mirror those from TruthfulQA: retaining only low-uncertainty predictions substantially improves accuracy, confirming that our uncertainty metrics can effectively filter unreliable reasoning traces.

## C. Additional Results on Mitigation

This appendix presents supplementary experimental results on UDPO-based mitigation, including adversarial robustness analysis and difficulty-stratified performance breakdown.

### C.1. Homogeneous Model Ensembles

Table 1 in the main text uses a heterogeneous ensemble of different LLM families. Here we evaluate homogeneous ensembles where all $N = 5$ agents use the same base model. This isolates the effect of model capacity from ensemble diversity.

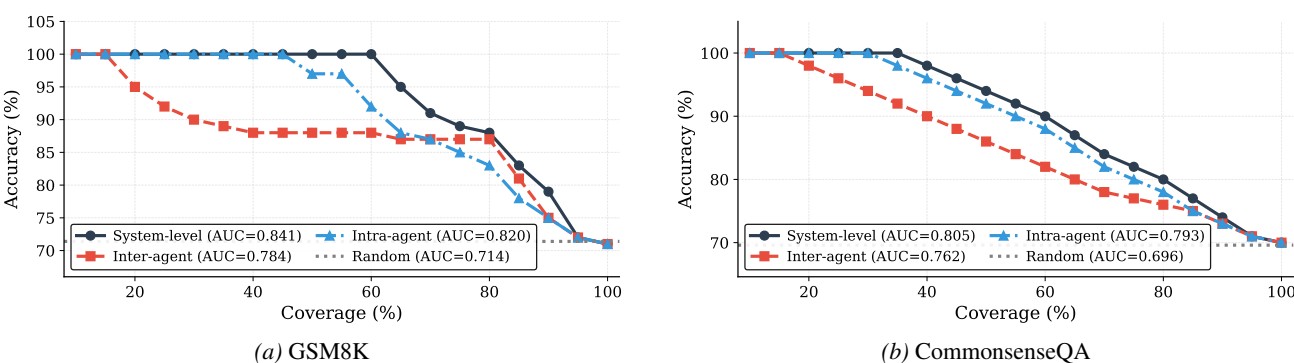

*(a)* GSM8K        *(b)* CommonsenseQA

*Figure 8.* **Selective prediction (additional datasets).** Supplements Figure 4 in the main text. Uncertainty-guided abstention consistently improves accuracy.

*Table 4.* **Homogeneous MAD results** with $N = 5$ agents of the same model type. We report accuracy (%) and uncertainty metrics. Compared to the heterogeneous ensemble in Table 1, homogeneous ensembles show slightly lower baseline performance but similar relative improvements from UDPO.

| Dataset | Model (5 agents) | Standard MAD | | | | MAPPO | | | | RMAAC | | | | UDPO (Ours) | | | |
|---|---|---|---|---|---|---|---|---|---|---|---|---|---|---|---|---|---|
| | | Acc | $U_{in}$ | $U_{ir}$ | $U_s$ | Acc | $U_{in}$ | $U_{ir}$ | $U_s$ | Acc | $U_{in}$ | $U_{ir}$ | $U_s$ | Acc | $U_{in}$ | $U_{ir}$ | $U_s$ |
| GSM8K | Llama-4 | 62.5 | .242 | .185 | .298 | 68.2 | .198 | .158 | .248 | 70.4 | .182 | .145 | .228 | **88.6** | **.072** | **.045** | **.068** |
| | Qwen-3 | 65.8 | .235 | .178 | .288 | 71.5 | .188 | .148 | .235 | 73.2 | .172 | .138 | .218 | **90.2** | **.068** | **.042** | **.062** |
| | Gemma-3n | 58.2 | .258 | .198 | .318 | 64.5 | .215 | .168 | .265 | 66.8 | .198 | .155 | .245 | **85.4** | **.078** | **.052** | **.075** |
| | Mixtral | 64.2 | .238 | .182 | .295 | 70.8 | .192 | .152 | .242 | 72.5 | .178 | .142 | .225 | **89.5** | **.070** | **.044** | **.065** |
| | Phi-4 | 60.5 | .248 | .192 | .308 | 66.8 | .205 | .162 | .255 | 68.5 | .188 | .148 | .235 | **86.8** | **.075** | **.048** | **.072** |
| TruthfulQA | Llama-4 | 66.2 | .228 | .155 | .248 | 70.5 | .185 | .132 | .212 | 72.8 | .168 | .125 | .198 | **86.5** | **.075** | **.052** | **.075** |
| | Qwen-3 | 69.5 | .218 | .145 | .238 | 73.8 | .172 | .122 | .198 | 75.5 | .158 | .115 | .185 | **88.2** | **.070** | **.048** | **.070** |
| | Gemma-3n | 62.8 | .245 | .172 | .272 | 67.2 | .198 | .148 | .232 | 69.5 | .182 | .138 | .215 | **83.8** | **.082** | **.058** | **.082** |
| | Mixtral | 68.2 | .222 | .148 | .242 | 72.5 | .178 | .125 | .205 | 74.2 | .162 | .118 | .192 | **87.5** | **.072** | **.050** | **.072** |
| | Phi-4 | 64.5 | .235 | .162 | .258 | 68.8 | .192 | .138 | .218 | 70.8 | .175 | .128 | .202 | **85.2** | **.078** | **.055** | **.078** |
| CSQA | Llama-4 | 70.2 | .212 | .148 | .238 | 74.5 | .168 | .125 | .202 | 76.2 | .155 | .118 | .188 | **88.5** | **.065** | **.042** | **.062** |
| | Qwen-3 | 73.5 | .202 | .138 | .225 | 77.2 | .158 | .115 | .188 | 78.8 | .145 | .108 | .175 | **90.2** | **.060** | **.038** | **.058** |
| | Gemma-3n | 66.8 | .228 | .165 | .258 | 71.2 | .185 | .138 | .218 | 73.5 | .168 | .128 | .202 | **85.8** | **.072** | **.048** | **.070** |
| | Mixtral | 72.2 | .208 | .142 | .232 | 76.5 | .162 | .118 | .195 | 78.2 | .148 | .112 | .182 | **89.5** | **.062** | **.040** | **.060** |
| | Phi-4 | 68.5 | .218 | .155 | .245 | 72.8 | .175 | .128 | .208 | 74.5 | .162 | .122 | .195 | **87.2** | **.068** | **.045** | **.065** |

**Observations.** Homogeneous ensembles generally achieve lower baseline accuracy than the heterogeneous ensemble (Table 1), as they lack the diversity benefits of combining different model architectures. However, UDPO provides consistent improvements across all model types, with accuracy gains of 18–24 percentage points over Standard MAD. Notably, Qwen-3 achieves the highest performance across all datasets, while Gemma-3n shows the largest relative improvement from UDPO, suggesting that our method is particularly effective for models with higher baseline uncertainty.

### C.2. Effect of Debate Rounds

The main experiments use $T = 5$ debate rounds. Here we evaluate the sensitivity to this choice by testing $T \in \{3, 5, 10\}$ with the heterogeneous ensemble ($N = 5$).

**Observations.** Increasing debate rounds improves all methods, but with diminishing returns for baselines. Standard MAD gains approximately 2–3% accuracy per additional 5 rounds, while UDPO gains 1.5–2%. Importantly, UDPO maintains a consistent advantage of 16–20 percentage points over Standard MAD regardless of $T$. The uncertainty metrics decrease with more rounds for all methods, indicating that extended deliberation allows agents to converge more reliably. However, the computational cost scales linearly with $T$, making $T = 5$ a practical default that balances performance and efficiency.

*Table 5.* **Effect of debate rounds** on performance with $N = 5$ agents. More rounds generally improve baseline methods but show diminishing returns; UDPO maintains substantial advantages across all settings.

| Dataset | Rounds $T$ | Standard MAD | | | | MAPPO | | | | RMAAC | | | | UDPO (Ours) | | | |
|---|---|---|---|---|---|---|---|---|---|---|---|---|---|---|---|---|---|
| | | Acc | $U_{in}$ | $U_{ir}$ | $U_s$ | Acc | $U_{in}$ | $U_{ir}$ | $U_s$ | Acc | $U_{in}$ | $U_{ir}$ | $U_s$ | Acc | $U_{in}$ | $U_{ir}$ | $U_s$ |
| GSM8K | $T$=3 | 64.2 | .248 | .195 | .312 | 69.5 | .205 | .165 | .258 | 71.2 | .188 | .152 | .238 | **89.5** | **.072** | **.048** | **.068** |
| | $T$=5 | 68.4 | .228 | .172 | .285 | 73.6 | .185 | .145 | .232 | 75.8 | .168 | .132 | .215 | **92.3** | **.065** | **.038** | **.058** |
| | $T$=10 | 70.5 | .215 | .158 | .268 | 75.2 | .172 | .132 | .218 | 77.5 | .158 | .122 | .202 | **93.8** | **.058** | **.032** | **.052** |
| TruthfulQA | $T$=3 | 67.5 | .238 | .165 | .262 | 70.8 | .195 | .142 | .225 | 72.5 | .178 | .132 | .208 | **85.8** | **.078** | **.058** | **.078** |
| | $T$=5 | 71.8 | .218 | .142 | .235 | 74.2 | .158 | .122 | .198 | 76.5 | .152 | .115 | .188 | **88.7** | **.068** | **.048** | **.068** |
| | $T$=10 | 73.2 | .202 | .128 | .218 | 76.5 | .145 | .108 | .182 | 78.2 | .138 | .102 | .172 | **90.5** | **.062** | **.042** | **.060** |
| CSQA | $T$=3 | 71.2 | .218 | .158 | .252 | 74.5 | .175 | .132 | .212 | 76.2 | .162 | .125 | .198 | **88.2** | **.068** | **.042** | **.062** |
| | $T$=5 | 75.4 | .198 | .135 | .225 | 78.2 | .148 | .112 | .188 | 79.8 | .142 | .108 | .178 | **91.5** | **.058** | **.032** | **.052** |
| | $T$=10 | 77.2 | .182 | .122 | .208 | 80.5 | .135 | .098 | .172 | 81.8 | .128 | .095 | .165 | **93.2** | **.052** | **.028** | **.045** |

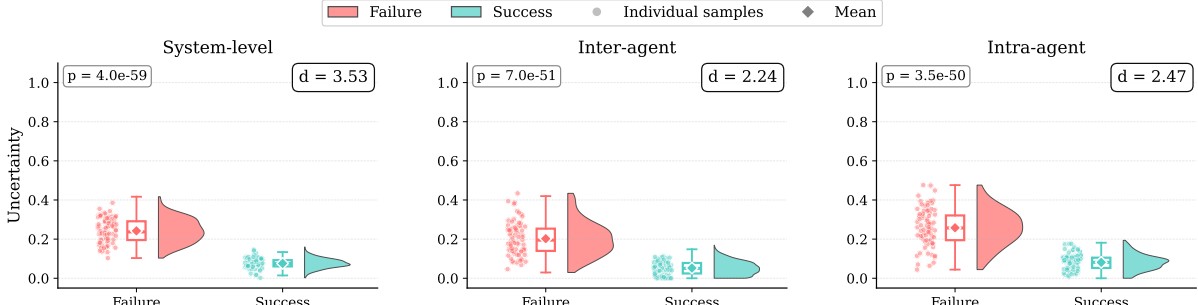

*Figure 9.* **Uncertainty distributions under adversarial attack (GSM8K).** Even with one malicious agent, incorrect predictions show significantly higher uncertainty across all metrics. Effect sizes (Cohen's $d$) remain substantial. Results on TruthfulQA and CommonsenseQA follow similar patterns.

## C.3. Adversarial Robustness: Mitigation Under Attack

This section evaluates whether our uncertainty metrics and UDPO mitigation remain effective when one agent is malicious.

### C.3.1. ATTACK SETUP

Following tse Huang et al. (2025), we use AUTOTRANSFORM to convert one debater into a malicious agent that introduces stealthy semantic errors—responses that appear logically sound but derive incorrect conclusions.

### C.3.2. UNCERTAINTY STILL SEPARATES CORRECT FROM INCORRECT

Under attack, incorrect predictions continue to exhibit substantially higher uncertainty than correct ones across all three metrics. Figure 9 illustrates this pattern on GSM8K, where effect sizes remain large (Cohen's $d = 2.24$–$3.53$). The separation is most pronounced for system-level uncertainty ($d = 3.53$), which aggregates signals from both intra-agent and inter-agent sources. We observe similar patterns on TruthfulQA and CommonsenseQA, confirming that our uncertainty decomposition retains its diagnostic power even when one agent actively undermines the debate.

### C.3.3. CORRELATIONS REMAIN STRONG

Figure 10 shows the correlation structure among uncertainty metrics and accuracy under attack on GSM8K. All three uncertainty types maintain strong negative correlations with accuracy ($r = -0.77$ to $-0.89$, all $p < 0.001$). The uncertainty metrics are also positively correlated with each other ($r = 0.57$–$0.83$), reflecting their shared sensitivity to debate instability. Table 6 summarizes these correlations and compares them to natural settings across datasets.

The interpretation is straightforward: when a malicious agent introduces errors, the other agents encounter arguments that conflict with their own reasoning but cannot easily identify the source of inconsistency. This confusion manifests as elevated uncertainty, which our metrics capture. The fact that uncertainty continues to predict failures under attack suggests that our

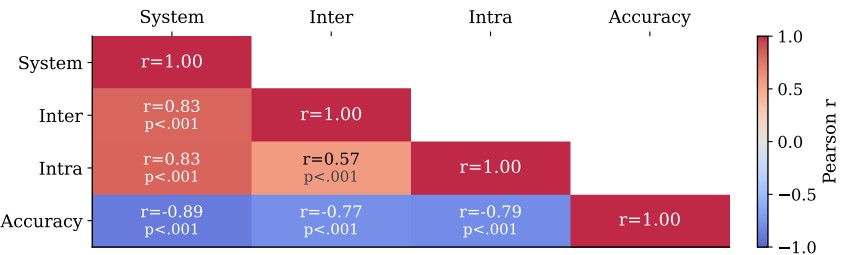

*Figure 10.* **Correlation matrix under adversarial attack (GSM8K).** Uncertainty metrics remain strongly negatively correlated with accuracy and positively correlated with each other. The pattern mirrors natural settings, indicating that adversarial perturbations do not disrupt the uncertainty–accuracy relationship.

*Table 6.* Correlation with accuracy under attack (Qwen). "Natural" shows correlations without attack for comparison.

| Metric | Attack | Natural | p-value |
|---|---|---|---|
| *GSM8K* | | | |
| $U_{\text{intra}}$ | -0.79 | -0.70 | $< 0.001$ |
| $U_{\text{inter}}$ | -0.77 | -0.68 | $< 0.001$ |
| $U_{\text{sys}}$ | -0.89 | -0.82 | $< 0.001$ |
| *TruthfulQA* | | | |
| $U_{\text{intra}}$ | -0.72 | -0.66 | $< 0.001$ |
| $U_{\text{inter}}$ | -0.75 | -0.71 | $< 0.001$ |
| $U_{\text{sys}}$ | -0.84 | -0.79 | $< 0.001$ |
| *CommonsenseQA* | | | |
| $U_{\text{intra}}$ | -0.68 | -0.62 | $< 0.001$ |
| $U_{\text{inter}}$ | -0.71 | -0.65 | $< 0.001$ |
| $U_{\text{sys}}$ | -0.81 | -0.76 | $< 0.001$ |

*Table 7.* Performance under attack (Qwen). "Drop" = change from natural setting.

| Method | GSM8K | Drop | TQA | Drop |
|---|---|---|---|---|
| Standard MAD | 35.2% | -34.6pp | 28.4% | -44.1pp |
| MAPPO | 41.7% | -32.5pp | 36.1% | -39.3pp |
| RMAAC | 52.3% | -23.8pp | 44.8% | -31.4pp |
| UDPO | 77.2% | -5.2pp | 66.4% | -14.8pp |

framework measures something fundamental about debate quality rather than artifacts specific to natural collapse.

Notably, the correlations under attack are slightly stronger than in natural settings across all datasets (e.g., $r = -0.89$ vs. $-0.82$ for $U_{\text{sys}}$ on GSM8K). This likely reflects that adversarial inputs create more extreme uncertainty values, sharpening the relationship between uncertainty and accuracy.

### C.3.4. UDPO IS MOST ROBUST

Table 7 compares performance degradation across methods. UDPO maintains the highest accuracy under attack (77.2% on GSM8K, 66.4% on TruthfulQA) and suffers the smallest performance drop from natural settings (-5.2pp and -14.8pp respectively). In contrast, Standard MAD loses over 34 percentage points, and even RMAAC—designed for robustness—drops by 23.8pp.

The robustness stems from explicit uncertainty decomposition: $U_{\text{intra}}$ detects reasoning inconsistencies forced by malicious inputs, $U_{\text{inter}}$ flags disagreement when stealthy errors conflict with correct reasoning, and $U_{\text{sys}}$ aggregates these signals into a natural fault-detection mechanism. Standard MAD lacks any defense; MAPPO's centralized coordination can be manipulated into false consensus; RMAAC assumes random noise rather than adversarial intent.

### C.4. Analysis by Question Difficulty

Figure 5 in the main text shows accuracy improvement stratified by $U_{\text{sys}}$. Here we present analogous analyses using $U_{\text{intra}}$ and $U_{\text{inter}}$ as stratification criteria to demonstrate the consistency of our findings.

**Observations.** The trends are consistent across all three uncertainty metrics:

• **Stratified by $U_{\text{intra}}$** (Figure 11a): UDPO improves from +2.3% (lowest level) to +17.1% (highest level). MAPPO shows +1.5% to +2.8%, peaking at +3.8% for medium uncertainty. RMAAC shows +1.9% to +3.2%, peaking at +4.8%.

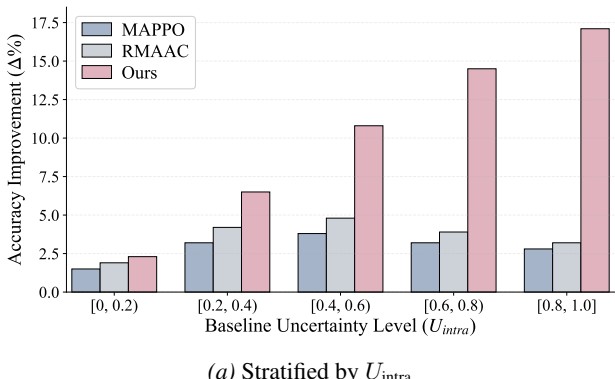

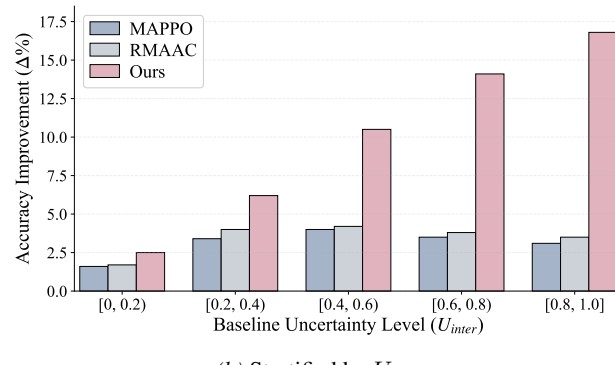

*(a)* Stratified by $U_{\text{intra}}$

*(b)* Stratified by $U_{\text{inter}}$

*Figure 11.* **Accuracy improvement ($\Delta\%$) by uncertainty level** using alternative stratification metrics. Supplements Figure 6 in the main text. Results averaged across all datasets and models. The pattern is consistent: UDPO shows monotonically increasing gains, while MAPPO and RMAAC plateau or decline at high uncertainty.

- **Stratified by $U_{\text{inter}}$** (Figure 11b): UDPO improves from +2.5% to +16.8%. MAPPO shows +1.6% to +3.1%, peaking at +4.0%. RMAAC shows +1.7% to +3.5%, peaking at +4.2%.

**Implications.** The consistency across all three stratification metrics confirms that our findings are robust. Regardless of which uncertainty metric defines "difficulty," UDPO provides larger gains on harder questions while both baselines plateau or decline. MAPPO and RMAAC both show an inverted-U pattern: they help most at medium difficulty and lose effectiveness at the extremes. This validates that UDPO fundamentally addresses debate collapse rather than exploiting artifacts of a particular metric.

## D. Uncertainty Quantification Details

This appendix provides formal definitions of the three components comprising the system-level uncertainty $U_{\text{sys}}$.

### D.1. Component Definitions

Let $\{a_T^{(1)}, \ldots, a_T^{(N)}\}$ denote the final-round responses from $N$ agents, and let $p(y \mid \tau)$ be the empirical distribution over candidate answers induced by these responses.

**Normalized Entropy.** The entropy component captures the spread of the answer distribution:

$$\tilde{H}(\tau) = -\frac{1}{\log K} \sum_{y \in \mathcal{Y}} p(y \mid \tau) \log p(y \mid \tau) \tag{16}$$

where $K = |\mathcal{Y}|$ is the number of distinct candidate answers. Normalization by $\log K$ ensures $\tilde{H}(\tau) \in [0, 1]$, with $\tilde{H}(\tau) = 0$ indicating unanimous agreement and $\tilde{H}(\tau) = 1$ indicating uniform distribution across all candidates.

**Disagreement Indicator.** The disagreement component indicates whether agents reach consensus:

$$D(\tau) = \mathbf{1}\left[\max_{y \in \mathcal{Y}} p(y \mid \tau) < 1\right] \tag{17}$$

This binary indicator equals 1 if any agent disagrees with the majority, and 0 only when all agents provide the same answer. While related to entropy, disagreement captures a distinct failure mode: even small minorities can indicate underlying reasoning conflicts.

**Leave-One-Out Instability.** The leave-one-out component measures the sensitivity of the final decision to individual agents:

$$L(\tau) = \frac{1}{N} \sum_{i=1}^{N} \mathbf{1}\left[\hat{y}(\tau) \neq \hat{y}(\tau^{-i})\right] \tag{18}$$

where $\hat{y}(\tau)$ is the majority-vote prediction using all $N$ agents, and $\hat{y}(\tau^{-i})$ is the majority-vote prediction after removing agent $i$. This component identifies fragile consensus: if removing a single agent changes the outcome, the collective decision lacks robustness.

### D.2. System-Level Uncertainty and Reward

The system-level uncertainty combines these three signals:

$$U_{\text{sys}}(\tau) = \frac{1}{3}\left[\tilde{H}(\tau) + D(\tau) + L(\tau)\right] \tag{19}$$

Each component captures a distinct aspect of collective unreliability: entropy measures answer dispersion, disagreement detects minority dissent, and leave-one-out instability identifies precarious majorities.

The corresponding reward used in UDPO (Section 4.2) directly inverts this uncertainty:

$$r_{\text{sys}}(\tau) = 1 - U_{\text{sys}}(\tau) = 1 - \frac{1}{3}\left[\tilde{H}(\tau) + D(\tau) + L(\tau)\right] \tag{20}$$

This reward is maximized when entropy is low (confident prediction), disagreement is absent (unanimous agreement), and leave-one-out instability is zero (robust consensus).

