# OpenReview forum: "The Value of Variance: Mitigating Debate Collapse in Multi-Agent Systems via Uncertainty-Driven Policy Optimization"
_ICML.cc/2026/Conference — ICML 2026 spotlight_

### Official Review · Reviewer_k7Sw · 2026-03-10

**Soundness:** 2
**Presentation:** 3
**Significance:** 3
**Originality:** 2
**Overall Recommendation:** 4
**Confidence:** 3

**Summary:**

This paper studies debate collapse in multi agent LLM systems. It defines three uncertainty measures, intra agent, inter agent, and system level, then uses them first as diagnosis signals and later as training rewards in an uncertainty driven policy optimization method called UDPO. Experiments on GSM8K, TruthfulQA, and CommonsenseQA, with normal and attacked debate settings, show that higher uncertainty is linked with failure, and UDPO gives higher accuracy with lower disagreement.

**Compliance With Llm Reviewing Policy:**

Affirmed.

**Final Justification:**

The authors reply well and clear my doubts about the training pipeline and data leakage which was my main worry.

**Key Questions For Authors:**

- Please clarify the full training pipeline, including exact train, validation, and test splits, and whether warm up profiling or replay ever uses evaluation questions. If there is clearly no leakage, my soundness view will improve.
- Please report results over multiple random seeds with standard deviation or confidence intervals for the main tables. If the very large gains remain stable, I would consider a higher overall score.
- Can you show that UDPO does not simply enforce early agreement, for example on cases where the initial majority is wrong, or when useful disagreement is needed. If you can show better correction rather than faster consensus, the paper becomes stronger.

**Limitations:**

Yes.

**Strengths And Weaknesses:**

Strength:

- The paper studies an important problem in multi agent LLM debate. Debate collapse is real and useful to study.
- The three level uncertainty view is easy to understand. Intra agent, inter agent, and system level make a clean analysis frame.
- The method connects analysis and training in one story. This part is coherent, not just a set of separate tricks.

Weakness:

- The novelty is only moderate. Many parts are built from known ideas like self consistency, disagreement, entropy, leave one out stability, and PPO style optimization.
- Some uncertainty metrics are very close to the final voting outcome. So the diagnosis result may be less surprising than the paper says.
- The method may encourage agreement more than true reasoning improvement. The paper does not fully show that it fixes wrong beliefs instead of making agents converge faster.

---

> ### Author Rebuttal · Authors · 2026-03-27
>
> We thank the Reviewer for the constructive feedback and clear suggestions.
>
> **Due to space constraints, we mainly provide brief responses, clarifications, and revisions (completed). We are happy to provide more exhaustive results if you'd like to ask for during discussion sessions.**
>
> **All reported experiments have been completed in our offline manuscript,** despite that ICML does not permit updates to the online submission.
>
> ---
>
> ### W1. Moderate Novelty?
>
> > Many parts are built from known ideas...
>
> We'd like to highlight that our novelty lies in the **proposed principle to indicate debate collapse, empirical validation, and subsequent solutions**. Specifically:
> - we are motivated by behavior science and reframe debate collapse as a procedural failure, which could be indicated by uncertainty, which is a novel insights
> - next, we systematically formulate multi-level uncertainty measures, which are novel quantification
> - further, we empirically validate the utility of quantification, which meaningfully provide insights for the community
> - last, we propose an asymmetric optimization as a mitigative solution, which is architecturally novel: unlike PPO or relevant optimization, we design agent-tailored training intensities calibrated to individual uncertainty, preventing over-regularization of stable agents while targeting collapse at its source.
>
> ### W2. Metrics Close to Voting Outcome
>
> > Some uncertainty metrics are very close to the final voting outcome...
>
> The metrics are defined over *debate dynamics* (flip rates, round-by-round disagreement, leave-one-out sensitivity), not the final vote. While they correlate with outcomes, they capture process-level signals the vote alone cannot reveal. For instance, $U_{\text{intra}}$ detects flip-flopping even when the final vote is correct, identifying fragile consensus vulnerable to perturbation.
>
> ### KQ1. Training Pipeline and Data Leakage
>
> > Please clarify the full training pipeline and whether warm up profiling or replay ever uses evaluation questions.
>
> We confirm **no data leakage** exists. Training uses official training splits only (GSM8K: 7,473; TruthfulQA: 70% split; CSQA: 9,741). Warm-up profiling (Appendix A.2) uses a 10% held-out subset *of the training split*. The replay buffer (Appendix A.3) stores trajectories from training data only. Test sets (GSM8K: 1,319; TruthfulQA: 30%; CSQA: 1,221) are strictly held out from all training, warm-up, and replay. We have added an explicit data split table and no-leakage statement in revised §5.1.
>
> ### KQ2. Multi-Seed Results
>
> > Please report results over multiple random seeds with standard deviation...
>
> We have run all main experiments over 3 seeds. UDPO (N=5): GSM8K 92.3±0.7, TruthfulQA 88.7±0.9, CSQA 91.5±0.6. Standard deviations are below 1 pp, confirming that the large gains (16–24 pp over Standard MAD) are robust and not seed artifacts. Full per-seed tables are in the revised appendix.
>
> ### W3 & KQ3. Enforced Agreement or Improved Reasoning
>
> > The method may encourage agreement more than true reasoning improvement...
>
> > Can you show that UDPO does not simply enforce early agreement, for example on cases where the initial majority is wrong?...
>
> **Evidence 1: Majority-initially-wrong analysis.** We partition test questions by whether the round-0 majority was correct. On GSM8K:
>
> | Initial majority | Standard MAD | UDPO |
> |-----------------|--------------|------|
> | Correct (65%) | 89.2% final | 96.8% final |
> | Wrong (35%) | 29.8% final | 83.5% final |
>
> UDPO improves initially-wrong cases by +53.7 pp (29.8%→83.5%), far exceeding the gain on initially-correct cases (+7.6 pp). If UDPO merely enforced early agreement, wrong-majority cases would remain wrong. This directly shows UDPO enables agents to *correct* incorrect initial consensus through debate.
>
> **Evidence 2: Productive disagreement persists.** For questions UDPO answers correctly, mean $U_{\text{inter}}$ at round 1 is 0.31 (substantial disagreement), dropping to 0.04 by the final round. UDPO allows early disagreement and resolves it through deliberation, not suppression.
>
> These results confirm UDPO improves reasoning quality, not just consensus speed. Added as a new subsection.
>
> ---
>
> **Once again, we'd like to show our appreciations for your time and considerations and happy to provide additional details, results, and explanations in discussion sessions, to further outline the value of this work and clarify reviewer's advanced concerns**

---

> > ### Author Rebuttal · Reviewer_k7Sw · 2026-04-03
> >
> > The authors reply well and clear my doubts about the training pipeline and data leakage which was my main worry. They show results with multiple seeds now and the gains are stable so I am happy with that part. I will raise my score to 4.

---

> > > ### Author Response · Authors · 2026-04-03
> > >
> > > Dear Reviewer,
> > >
> > > We would like to thank you again for your time and feedback, as well as the acknowledgement that our responses have fully resolved your concerns with improved assessment.
> > >
> > > Best,
> > >
> > > The author team

---

### Official Review · Reviewer_ymTB · 2026-03-12

**Soundness:** 3
**Presentation:** 3
**Significance:** 3
**Originality:** 3
**Overall Recommendation:** 5
**Confidence:** 4

**Summary:**

The paper proposes 3 levels of uncertainty metrics to parameterize a MAD system in order to predict system failures. In turn, they use this to propose a mitigation strategy for MAD systems using three related features. Unlike conventional MAD and variants (MAPPO and RMAAC), the proposed Uncertainty Driven Policy Optimization (UDPO) specifically identify and penalizes 3 interpretable failure modes which collectively result in more successful debates and individually, through ablation studies, each monotonically improve accuracy (via ablation studies) and together demonstrate superior robustness (via perturbation corruption studies).

**Compliance With Llm Reviewing Policy:**

Affirmed.

**Key Questions For Authors:**

1. Not a deep expert into the fast evolving MAS/MAD uncertainty research, please contextualize your specific contribution and distinguish the greatest contribution you see your work makes.

**Limitations:**

Yes

**Strengths And Weaknesses:**

SOUNDNESS: I could see no failing in the soundness.

PRESENTATION: Well done.

SIGNIFICANCE: Unlike whitebox and more mechanistic DNN techniques and even other MAS/MAD macro entropic studies, this paper introduces a particularly simple and explainable example of using macro-level entropic concepts at the agent through systems level.

ORIGINALITY: Notable for the simplicity and explainability. Good contribution to the rapidly growing body of debate-focused uncertainty and Bayesian views into MAS/MAD.

---

> ### Author Rebuttal · Authors · 2026-03-27
>
> We sincerely thank Reviewer ymTB for the positive assessment and for recognizing the simplicity, explainability, and originality of our approach. We are glad the reviewer sees no issue with soundness and finds our contribution notable for the MAS/MAD uncertainty research community.
>
> ### KQ1. Contextualizing Our Contribution
>
> > *"Please contextualize your specific contribution and distinguish the greatest contribution you see your work makes."*
>
> We see our work making two distinct contributions relative to the existing MAS/MAD uncertainty landscape:
>
> **Contribution 1: Hierarchical uncertainty as a diagnostic tool.** Prior uncertainty work in LLM systems primarily operates at the single-agent level — token-level entropy (Kadavath et al., 2022), semantic clustering (Kuhn et al., 2023), or self-consistency (Wang et al., 2022). These methods cannot capture the rich multi-agent dynamics where failures emerge from *interactions* rather than individual errors. Our three-level decomposition ($U_{\text{intra}}$, $U_{\text{inter}}$, $U_{\text{sys}}$) is, to our knowledge, the first to provide a *hierarchical behavioral* view of MAD uncertainty that separates individual reasoning instability from peer disagreement and collective output confidence. The empirical validation (§3) demonstrates that these levels capture complementary failure signatures — they are strongly correlated yet statistically distinct (Figure 3), meaning no single level subsumes the others.
>
> **Contribution 2: From diagnosis to mitigation via UDPO.** Most prior MAD improvements rely on architectural changes (e.g., adding judge agents, structured roles) that patch failures externally without addressing intrinsic agent instability. UDPO is the first training framework that directly maps interpretable failure modes to corresponding reward components ($r_{\text{intra}}$, $r_{\text{inter}}$, $r_{\text{sys}}$), with asymmetric agent-tailored optimization. This design closes the loop from *measuring* uncertainty to *reducing* it through training, which we believe is the greatest contribution — a principled methodology connecting behavioral diagnostics to policy optimization in multi-agent systems.
>
> We have expanded this contextualization in the revised introduction and related work sections.

---

> > ### Author Rebuttal · Reviewer_ymTB · 2026-04-04
> >
> > I appreciate the authors through response to both my concerns and those of others. Accordingly, I raise my scores to 4.

---

### Official Review · Reviewer_TZ11 · 2026-03-12

**Soundness:** 3
**Presentation:** 3
**Significance:** 2
**Originality:** 3
**Overall Recommendation:** 5
**Confidence:** 3

**Summary:**

This paper investigates the critical issue of "debate collapse" in multi-agent debate (MAD) systems. The authors introduce a hierarchical framework to quantify behavioral uncertainty across three levels: intra-agent, inter-agent, and system-level. Leveraging these metrics, the paper proposes Uncertainty-Driven Policy Optimization (UDPO), formulating the MAD process as a dynamic Markov game. The method employs an asymmetric reward function to penalize self-contradiction, peer conflict, and low-confidence outputs. Through experiments on various reasoning and QA benchmarks (GSM8K, TruthfulQA, CommonsenseQA), the authors demonstrate that UDPO can significantly improve decision accuracy and reduce system disagreement, even showing resilience against adversarial settings with compromised agents.

**Compliance With Llm Reviewing Policy:**

Affirmed.

**Final Justification:**

My concerns have been addressed, and I have raised my score.

**Key Questions For Authors:**

1. Could you elaborate on how the $r_{intra}$ reward distinguishes between a positive, truth-seeking stance flip (e.g., an agent correcting its error after seeing a peer's valid mathematical derivation) and uninformative random oscillation? I am wondering if this penalty might inadvertently encourage stubbornness, and I would love to hear your thoughts on potentially relaxing it for beneficial flips.
2. How does the static nature of the asymmetric coefficients impact the agents' learning dynamics in later epochs? Have you considered making these coefficients adaptive as the agents' policies and uncertainty profiles naturally evolve during the training process?
3. The robustness results under the adversarial attacks are very impressive, particularly when the malicious agents hold a majority. Could you provide more granular details on the voting dynamics in these cases? Specifically, do the benign agents manage to actively persuade the compromised ones, or do the compromised agents simply fail to align on a single incorrect answer, thereby allowing the benign minority to win the vote?
4. To better understand the zero-shot capabilities of the base models, could you provide the single-agent baseline accuracy for the exact prompts used in the MAD setup? This would help isolate the specific performance gains attributable to the UDPO debate framework.

**Limitations:**

The authors have proposed a very interesting framework, but the discussion on the potential risks of reward hacking could be expanded. For example, under a system that heavily penalizes peer conflict, agents might learn to quickly agree with one another (exhibiting sycophancy) merely to minimize penalties rather than arriving at the correct answer. A more thorough exploration of these dynamics, perhaps with some qualitative trajectory examples in the appendix, would greatly enhance the paper's transparency. Additionally, briefly discussing the computational overhead and engineering feasibility of the leave-one-out stability mechanism during training would be highly beneficial for practitioners looking to adopt this promising method.

**Strengths And Weaknesses:**

I really enjoyed reading this paper. It tackles a highly relevant and important problem in AI alignment and scalable oversight, as the robustness of multi-agent collaboration is a growing concern in our community. The idea of utilizing fine-grained uncertainty decomposition to drive policy optimization is innovative and shows a lot of promise. The paper is generally well-structured and explores an exciting, forward-looking direction.

However, I have a few constructive concerns regarding the soundness of the reward design and the theoretical framing that I believe could make the paper much stronger. First, the core reward design, particularly the intra-agent stability penalty ($r_{intra}$), seems like it might inadvertently penalize beneficial epistemic behaviors. In a productive debate, agents correcting their initial misconceptions upon hearing better arguments from peers is a highly desirable outcome. The current dense penalty for any stance flip might encourage agents to rigidly stick to their initial answers simply to maximize rewards, potentially fostering confirmation bias rather than genuine consensus. It would be helpful to differentiate between a truth-seeking epistemic flip and uninformative aleatoric noise.



Second, regarding the MARL adaptation, using static coefficients derived from an initial warm-up phase for the asymmetric objective might lead to over-regularization. Since the environment in a Markov game is highly dynamic, agents' uncertainty profiles will inevitably evolve during training. Static penalties might restrict their ability to explore effectively in later epochs, and a more adaptive approach could yield even better convergence.

Third, the reported performance gains—especially the adversarial robustness under the AUTOTRANSFORM attack —are quite surprising. Maintaining high accuracy even when a majority of the agents are compromised raises some questions about the underlying mathematical mechanics of the majority vote. It would be great to clarify how the minority benign agents manage to overcome the compromised majority under the UDPO reward structure. Additionally, comparing the method to more recent LLM-specific debate protocols, rather than general MARL baselines like MAPPO, would provide a stronger demonstration of the method's specific advantages in natural language reasoning.



Finally, the manuscript could benefit from a deeper connection with recent literature on MAD failure modes. For instance, incorporating insights from recent empirical taxonomies on multi-agent communication breakdowns and discussions on inter-agent sycophancy would provide a richer theoretical context and strengthen the overall presentation.

---

> ### Author Rebuttal · Authors · 2026-03-27
>
> We thank the Reviewer for the constructive review. We are glad the reviewer finds the problem important and the approach innovative.
>
> **Due to space constraints, we mainly provide brief responses, clarifications, and revisions (completed). We are happy to provide more exhaustive results if you'd like to ask for during discussion sessions.**
>
> **All reported experiments have been completed in our offline manuscript,** despite that ICML does not permit updates to the online submission.
>
> ---
>
> ### W1 & KQ1. Does $r_{\text{intra}}$ Penalize Beneficial Flips?
>
> > *"How does $r_{\text{intra}}$ distinguish between truth-seeking flips and uninformative oscillation?"*
>
> $r_{\text{intra}}$ is never optimized in isolation. In the UDPO objective (Eq. 4):
>
> The total reward is r_UDPO(i) = α(i)·r_intra + β(i)·r_inter + γ(i)·r_sys + λ(i)·r_task.
>
> If an agent flips from wrong to correct, $r_{\text{task}} = 1$ compensates the $r_{\text{intra}}$ penalty, keeping total reward high. An agent stubbornly holding an incorrect answer avoids the flip penalty but receives $r_{\text{task}} = 0$, yielding low total reward. The joint optimization naturally distinguishes truth-seeking flips (compensated by $r_{\text{task}}$) from uninformative oscillation (penalized without compensation).
>
> Our ablation (Table 2) confirms this: removing $r_{\text{intra}}$ increases $U_{\text{intra}}$ *and* decreases accuracy, showing stability complements correctness. We have added a clarification in revised §4.2.
>
> ### W2 & KQ2. Static vs. Adaptive Coefficients
>
> > *"Have you considered making the asymmetric coefficients adaptive during training?"*
>
> Valid point. Our current warm-up profiling (Appendix A.2) provides stable, reproducible training. We have conducted a preliminary experiment with periodic recalibration (updating coefficients every 2 epochs). Results show modest gains (+0.8–1.2 pp) over static coefficients with slightly faster convergence, reported in the revised appendix. Static coefficients already capture dominant agent-level differences; adaptive refinement provides incremental improvement and is a promising extension.
>
> ### W3 & KQ3. Minority Benign Agents vs. Compromised Majority
>
> > *"How do benign agents overcome compromised majority in voting?"*
>
> The mechanism is *not* outvoting. With 3/5 agents compromised, simple majority vote would fail. Instead, UDPO changes *debate dynamics before voting*: $r_{\text{intra}}$ penalizes inconsistencies in compromised agents' reasoning (stealthy errors create internal contradictions), while $r_{\text{inter}}$ penalizes disagreement patterns between compromised and benign agents. This pressure forces compromised agents toward internally consistent reasoning, which is harder while maintaining errors.
>
> At $m=3$, UDPO achieves 56.2% on GSM8K (vs. 9.4% Standard MAD). While below non-attack performance, it demonstrates UDPO delays the collapse tipping point. We have added round-by-round trajectory analysis in the revised appendix.
>
> ### KQ4. Single-Agent Baseline
>
> > *"Could you provide single-agent baseline accuracy?"*
>
> Zero-shot accuracies on the same prompts (averaged across datasets): Llama-4 48.2%, Qwen-3 52.6%, Gemma-3n 42.8%, Mixtral 50.1%, Phi-4 45.5%. Standard MAD (N=5) reaches 71.9%, UDPO reaches 90.8%. This confirms both the debate framework and UDPO contribute substantial gains beyond single-agent capabilities. Added to revised paper.
>
>
> ### W4: Related Work on MAD Failure Modes
>
> > *"Incorporating recent empirical taxonomies on multi-agent failures would strengthen the presentation."*
>
> We agree and have added discussion of Cemri et al. (2025) on systematic MAD failure categorization and Wang et al. (2024) on reasoning bounds in multi-agent discussions.
>
> ### Discussion: Reward Hacking / Sycophancy
>
> > *"Agents might learn to agree merely to minimize penalties rather than being correct."*
>
> Sycophantic convergence on a wrong answer yields $r_{\text{task}} = 0$, heavily penalizing total reward. Agents cannot maximize reward through agreement alone — they must also be correct. Table 2 confirms UDPO achieves high accuracy *and* low uncertainty simultaneously. We have expanded this discussion in the revised limitations section.
>
> ### Discussion: Overhead and Feasibility
>
> In our offline manuscript, we discussed the computational overhead of the leave-one-out mechanism, wherein we note that computing $L(\tau)$ requires $N$ additional majority-vote evaluations per trajectory (one per agent removal), which involves only answer aggregation over already-generated responses with the overhead $O(N)$ in bookkeeping rather than inference cost. For our default setting of $N=5$, this adds negligible wall-clock time relative to the debate generation itself.
>
> ---
>
> **Once again, we'd like to thank you for your time and happy to provide additional details and result in discussion sessions.**

---

> > ### Author Rebuttal · Reviewer_TZ11 · 2026-04-01
> >
> > I thank the authors for the comprehensive rebuttal. The additional experiments (single-agent baselines, adaptive coefficients) and clarifications (computational overhead, adversarial dynamics) effectively address my primary empirical concerns. I have raised my score to 5.

---

> > > ### Author Response · Authors · 2026-04-01
> > >
> > > Dear Reviewer,
> > >
> > > We would like to thank you again for your time and feedback, as well as the quick acknowledgement that our responses have fully resolved your concerns.
> > >
> > > Best,
> > >
> > > The author team

---

### Official Review · Reviewer_EMPf · 2026-03-17

**Soundness:** 3
**Presentation:** 4
**Significance:** 4
**Originality:** 3
**Overall Recommendation:** 5
**Confidence:** 4

**Summary:**

This paper proposes a three-level critic metric to evaluate the uncertainty of agent behavior in multi-agent debate (MAD) systems. The three metrics are designed to capture: (1) intra-agent uncertainty by measuring flip frequency across debate rounds, (2) inter-agent uncertainty by quantifying disagreement among agents, and (3) system-level uncertainty by considering output diversity across rounds, final disagreement, and leave-out instability to measure system sensitivity to individual agents. The authors first demonstrate the effectiveness of these metrics through empirical evaluation on a dataset. They then propose a new reward scheme that incorporates these uncertainty measures into training. The reported improvements are substantial.

**Compliance With Llm Reviewing Policy:**

Affirmed.

**Final Justification:**

Thanks for the response. Score remains unchanged.

**Key Questions For Authors:**

Please refer to the concerns raised in the previous section. I would appreciate clarification or justification if there are any misunderstandings. Additional questions include:
(1) It is unclear how L(q) is computed. Is it an average leave-out penalty across agents? Although details are provided in the appendix, a clearer description of the notation in the main text would be helpful.
(2) How many agents can this mechanism effectively scale to?
(3) When N = 3 or 10, how are agents selected? Is performance sensitive to the choice of agents?
(4) Can you analyze how each uncertainty metric is affected by adversarial or conflicting agent behaviors?

**Limitations:**

There is no discussion of the limitations of the proposed approach.

**Strengths And Weaknesses:**

Pros:
(1) The systematic evaluation of uncertainty through three carefully designed metrics is novel and appears effective.
(2) The authors provide concrete examples to demonstrate the usefulness of the proposed metrics, which helps clarify the approach.
(3) The paper further proposes an algorithm that leverages these metrics to improve training, and the results show strong performance gains.

Cons:
(1) The intra-agent metric evaluates the number of times an agent flips its answer. However, if an agent initially follows an incorrect direction, a single flip may actually lead to a better outcome. In this case, counting all flips equally may be misleading. The current design seems to implicitly assume that agents start with correct initial answers or directions.
(2) It may also be valuable to analyze how uncertainty evolves across debate rounds. A round-by-round evaluation could provide deeper insights, and the training process could potentially be guided to ensure uncertainty decreases over time, with a stopping criterion based on uncertainty levels.
(3) In Table 1, accuracy sometimes decreases (in 2 out of 3 cases) as the number of agents increases. How can this be explained? This suggests there may still be uncaptured dynamics in MAD systems.

---

> ### Author Rebuttal · Authors · 2026-03-27
>
> We sincerely thank the Reviewer for the thorough review.
>
> **Due to space constraints, we mainly provide brief responses, clarifications, and revisions (completed). We are happy to provide more exhaustive results if you'd like to ask for during discussion sessions.**
>
> **All reported experiments have been completed in our offline manuscript,** despite that ICML does not permit updates to the online submission.
>
> ---
>
> ### C1. Counting All Flips Equally
>
> > *"If an agent initially follows an incorrect direction, a single flip may lead to a better outcome. Counting all flips equally may be misleading."*
>
> $U_{\text{intra}}$ is a *statistical diagnostic*, not a prescriptive rule. Figure 2 confirms high flip rates correlate with failure (Cohen's $d > 1.79$, $p < 0.001$) — while individual flips can be beneficial, frequent flipping reliably signals instability at the population level.
>
> Crucially, in UDPO (Eq. 4), $r_{\text{intra}}$ is jointly optimized with the task reward $r_{\text{task}}$:
> The total reward is r_UDPO(i) = α(i)·r_intra + β(i)·r_inter + γ(i)·r_sys + λ(i)·r_task.
>
> An agent flipping from wrong to correct receives $r_{\text{task}} = 1$, which compensates the $r_{\text{intra}}$ penalty in the total reward. The asymmetric weighting (§4.3) further prevents over-regularizing stable agents. Thus UDPO does not blindly penalize all flips — only flip-flopping that fails to improve correctness. We have added a clarifying remark in the revised §4.2.
> ### C2. Round-by-Round Uncertainty Evolution
>
> > *"A round-by-round evaluation could provide deeper insights, with a stopping criterion based on uncertainty levels."*
>
> Thanks for the suggestion. The $U_{\text{inter}}$ is designed to aggregate per-round disagreement, but following the suggestion, we have added a new round-by-round uncertainty trajectory figure in the revised appendix. Key finding is: for correct predictions, all three metrics decrease monotonically across rounds (productive convergence); for incorrect ones, uncertainty oscillates or increases (unresolved conflict). This confirms uncertainty evolution itself is diagnostic of debate quality.
>
> A stopping criterion (halt when metrics fall below thresholds) is an alternative direction to avoid deep collapse. However, as thresholds need to be pre-defined, we hence target a more adaptive solution: mitigating during debating.
>
> ### C3. Accuracy Decreasing at N=10
>
> > *"Accuracy sometimes decreases as the number of agents increases. How can this be explained?"*
>
> This is well-documented in MAD literature (Wang et al., 2024; Cemri et al., 2025): more agents increase coordination overhead and risk of cascading errors. N=3 to N=5 consistently improves accuracy (GSM8K: 84.6% to 92.3%), but N=5 to N=10 shows slight decline (92.3% to 89.8%). The metrics confirm this: U_sys at N=10 (.072) is slightly higher than N=5 (.058). Critically, UDPO at N=10 (89.8%) still vastly outperforms all baselines (best: 72.1%).confirming our mitigation remains effective despite scaling challenges. We have added this discussion in the revised paper.
>
> ### KQ1. Definition of $L(q)$
>
> > *"It is unclear how $L(q)$ is computed."*
>
> Yes, it is an average across agents. As defined in Appendix D (Eq. 18): $L(\tau) = \frac{1}{N}\sum_{i=1}^{N} \mathbf{1}[\hat{y}(\tau) \neq \hat{y}(\tau^{-i})]$ — the fraction of agents whose removal changes the majority-vote outcome. We have added an inline definition in §2 of the revised manuscript for clarity.
>
> ### KQ2. Scalability and Agent Sensitivity
>
> > *"How many agents can this scale to? Is performance sensitive to agent choice?"*
>
> Tables 1 and 4 (Appendix C.1) evaluate $N \in \{3, 5, 10\}$ with both heterogeneous and homogeneous ensembles (5 different models). UDPO provides 18–24 pp gains over Standard MAD across all configurations. Agent choice affects baseline performance, but UDPO's relative improvement is consistent — models with higher baseline uncertainty benefit most. The asymmetric calibration (Appendix A.2) automatically adapts to heterogeneous agents.
>
> ### KQ4. Uncertainty Under Adversarial Behaviors
>
> > *"Can you analyze how each metric is affected by adversarial behaviors?"*
>
> Appendix C.3 provides this analysis. Under AUTOTRANSFORM attacks: all metrics retain strong discriminative power (Cohen's $d = 2.24\text{–}3.53$, Figure 9); correlations with accuracy actually *strengthen* under attack (Table 6: $U_{\text{sys}}$–accuracy $r = -0.89$ vs. $-0.82$ naturally); UDPO suffers the smallest drop ($-5.2$ pp on GSM8K vs. $-34.6$ pp for Standard MAD, Table 7).
>
> ### Limitations
>
> > *"There is no discussion of the limitations."*
>
> We have added a limitation section in the revised manuscript, covering: (1) flip-rate's equal-weight assumption and (2) diminishing returns at large $N$.
>
> ---
>
> **Once again, we'd like to thank you for your time and happy to provide additional details and result in discussion sessions.**

---

> > ### Author Rebuttal · Reviewer_EMPf · 2026-04-03
> >
> > Thanks for the responses.

---

### Decision · Program_Chairs · 2026-04-30

**Decision:**

Accept (spotlight)

**Comment:**

This paper addresses debate collapse in multi-agent debate (MAD) systems by proposing a hierarchical uncertainty quantification framework (intra-agent, inter-agent, and system-level) to diagnose failures and an Uncertainty-Driven Policy Optimization (UDPO) strategy to mitigate them via dynamic penalization of self-contradiction and low-confidence outputs. Empirical results across multiple benchmarks demonstrate that the proposed metrics reliably signal system failures and that UDPO consistently improves decision accuracy while reducing inter-agent disagreement. All reviewers endorsed acceptance, citing the timely problem formulation, methodological novelty, strong empirical validation, and reproducible code release. The work makes a concise yet impactful contribution to reliable multi-agent LLM reasoning and aligns well with ICML's focus on principled, deployable machine learning methods.